



# Analysis of $CO_2$, $CH_4$ and CO surface and column concentrations observed at Reunion Island by assessing WRF-Chem simulations

Sieglinde Callewaert[1], Jérôme Brioude[2], Bavo Langerock[1], Valentin Duflot[2], Dominique Fonteyn[1], Jean-François Müller[1], Jean-Marc Metzger[3], Christian Hermans[1], Nicolas Kumps[1], Emmanuel Mahieu[4], and Martine De Mazière[1]

[1]Royal Belgian Institute for Spacy Aeronomy (BIRA-IASB), Brussels, Belgium
[2]Laboratoire de l'Atmosphère et des Cyclones (LACy), UMR8105, Saint-Denis, Reunion Island, France
[3]UAR 3365 - OSU Réunion, Université de La Réunion, Saint-Denis, Reunion Island, France
[4]UR SPHERES, Department of Astrophysics, Geophysics and Oceanography, University of Liège, Liège, Belgium

**Correspondence:** Sieglinde Callewaert (sieglinde.callewaert@aeronomie.be)

**Abstract.** Reunion Island is situated in the Indian Ocean and holds one of the very few atmospheric observatories in the tropical Southern Hemisphere. Moreover, it hosts experiments providing both ground-based surface and column observations of $CO_2$, $CH_4$ and CO atmospheric concentrations. This work presents a comprehensive study of these observations made in the capital Saint-Denis and at the high-altitude Maïdo Observatory. We used simulations of the Weather Research and Forecasting model coupled with Chemistry (WRF-Chem), in its passive tracer option (WRF-GHG), to gain more insight in the factors that determine the observed concentrations. Additionally, this study provides an evaluation of the WRF-GHG performance in a region of the globe where it has not yet been applied.

A comparison of the basic meteorological fields near the surface and along atmospheric profiles showed that WRF-GHG has decent skill in reproducing these meteorological measurements, especially temperature. Furthermore, a distinct diurnal $CO_2$ cycle with values up to 450 ppm was found near the surface in Saint-Denis, driven by local anthropogenic emissions, boundary layer dynamics and accumulation due to low wind speed at night. Due to an overestimation of local wind speed, WRF-GHG underestimates this nocturnal buildup. At Maïdo, a similar diurnal cycle is found but with much smaller amplitude. There, surface $CO_2$ is essentially driven by the surrounding vegetation. The hourly column-averaged mole fractions of $CO_2$ ($XCO_2$) of WRF-GHG and the corresponding TCCON observations were highly correlated with a coefficient of 0.90. These observations represent different air masses than those near the surface, they are influenced by processes from Madagascar, Africa and further away. The model shows contributions from fires during the Southern Hemisphere biomass burning season, but also biogenic enhancements associated with the dry season. Due to a seasonal bias in the boundary conditions, WRF-GHG fails to accurately reproduce the $CH_4$ observations at Reunion Island. Further, local anthropogenic fluxes are the largest source influencing the surface $CH_4$ observations. However, these are likely overestimated. Further, WRF-GHG is capable of simulating CO levels on Reunion Island with a high precision. As to the observed CO column (XCO), we confirmed that biomass burning plumes from Africa and elsewhere are important for explaining the observed variability. The in situ observations at the Maïdo Observatory can characterize both anthropogenic signals from the coastal regions and biomass burning enhancements from afar. Finally, we found that a high model resolution of 2 km is needed to accurately represent the surface observations. At Maïdo an even


higher resolution might be needed because of the complex topography and local wind patterns. To simulate the column FTIR
observations on the other hand, a model resolution of 50 km might already be sufficient.

# 1 Introduction

Major greenhouse gases such as carbon dioxide ($CO_2$) and methane ($CH_4$) have a direct impact on the radiative forcing of
the atmosphere. They are the main drivers of climate change, since their global mean concentrations have increased over the
industrial era by about 47 % and 156 %, for $CO_2$ and $CH_4$ respectively, as a result of human activities (Masson-Delmotte
et al., 2021). Carbon monoxide (CO) on the other hand is not a greenhouse gas but indirectly affects the lifetime of $CH_4$ in the
atmosphere through its competing reaction with OH. Additionally it plays a major role in air pollution as it participates in the
formation of tropospheric ozone and urban smog.

The importance of these gases, hereafter all referred to as greenhouse gases (GHG), has led to the establishment of global
observation networks to monitor their trends and variability. Ground-based remote sensing networks such as the Network for
the Detection of Atmospheric Composition Change (NDACC) and the Total Carbon Column Observing Network (TCCON)
are known for their long time series of accurate column observations (De Mazière et al., 2018; Wunch et al., 2011). The
Fourier Transform infrared (FTIR) spectrometer observations carried out in these networks use direct sunlight to measure the
absorption of atmospheric trace gases along the line-of-sight and provide precise information on the total column abundance
or vertical profile of GHG and other species. They are used by scientists worldwide to detect changes in the atmospheric
composition, to improve our understanding of the carbon cycle, to provide validation for space-based measurements, or to trace
down emissions. In addition to FTIR observations, surface in situ observations of these gases are carried out to better constrain
sources and sinks on a smaller, more local scale. Both observation types contain valuable information on the emissions and
transport of these species and are complementary.

Reunion Island (55 °E, 21 °S) is a French island in the Indian Ocean, situated about 550 km east of Madagascar. It hosts
one of the very few atmospheric observatories in the tropical Southern Hemisphere, which provides both ground-based in
situ and FTIR observations of GHG, contributing to the Integrated Carbon Observation System (ICOS) and NDACC and
TCCON, respectively. GHG observations at Reunion Island are made at two sites: in the capital Saint-Denis and at the high-
altitude Maïdo Observatory (Baray et al., 2013). Several studies already investigated the factors influencing the observations
at Reunion Island. Zhou et al. (2018) analyzed the trends and seasonal cycles of $CH_4$ and CO by comparing the ground-
based remote sensing and in situ observations. They noticed a distinct seasonal cycle in the column-averaged dry-air mole
fractions of CO (XCO) with peak values between September and November, linked to the biomass burning season in Africa
and South America, which confirmed earlier work from Duflot et al. (2010). Furthermore, backward trajectory simulations
revealed different origins of air masses observed at Reunion Island near the surface and higher up, resulting in surface CO
concentrations that are systematically lower than XCO. Near the surface, winds generally originate in the Indian Ocean, while
higher up the air comes from Africa and South America. The ability to detect biomass burning plumes at Reunion Island was
also reported by Vigouroux et al. (2012). The available $XCO_2$ timeseries has however not yet been investigated. Additionally,





the Maïdo Observatory hosts a wide range of instruments of which the measurements have already been used by a variety of scientists to characterize the processes that occur at this particular location (Guilpart et al., 2017; Foucart et al., 2018; Duflot et al., 2019; Verreyken et al., 2021). However the in situ observations at the Maïdo Observatory of the longer-lived species

$CO_2$ and $CH_4$ have not yet been studied in detail, and this applies also to the available surface measurements at Saint-Denis. Therefore the aim of the current work is to make a comprehensive description and analysis of in situ and column observations of $CO_2$, $CH_4$ and CO at Reunion Island, both at Saint-Denis and Maïdo. To gain more insight into the factors that influence the observed concentrations, we will rely on simulations of the widely used Weather Research and Forecasting model coupled with chemistry (WRF-Chem; Skamarock et al. (2008)), in its passive tracer option called WRF-GHG (Beck et al., 2013). This

regional atmospheric model simulates 4D fields of $CO_2$, $CH_4$ and CO, resulting from their sources, sinks and transport in the troposphere, without interaction with other species, while accounting for the meteorology. It makes it possible to separate each chemical compound into several tracers representing the contributions of different emissions sources within the model domain such as anthropogenic, biogenic, biomass burning etc... Moreover it supports the online calculation of biogenic $CO_2$ fluxes following the Vegetation Photosynthesis and Respiration Model (VPRM; Mahadevan et al. (2008)). Thus far, applications of

WRF-GHG have mainly focused on $CO_2$ to study city emissions (Pillai et al., 2016; Feng et al., 2016; Park et al., 2018; Zhao et al., 2019), or to evaluate the VPRM model (Ahmadov et al., 2007; Jamroensan, 2013; Dayalu et al., 2018; Hu et al., 2020; Park et al., 2020). It has also been used in combination with in situ and column observations, flux towers and satellite measurements to better understand the carbon cycle (Pillai et al., 2010, 2012; Liu et al., 2018; Li et al., 2020). The model was shown to be an excellent tool for studying regional carbon budgets and is therefore very well suited to our needs. Few studies

have used WRF-GHG to simulate $CH_4$ and CO, and these studies focused on explaining enhancements identified by satellite instruments (Beck et al., 2013; Dekker et al., 2017; Tsivlidou, 2018; Borsdorff et al., 2019; Dekker et al., 2019; Verkaik, 2019). Hence, this work additionally aims at evaluating the model performance for these species in a region where it has not yet been applied. This might potentially draw attention to shortcomings in the model, allowing and motivating the model community to improve it.

This manuscript focuses on the factors that influence the observed GHG concentrations and their variations at Saint-Denis and the Maïdo Observatory. In particular, it addresses the following questions: (1) To what extent are the observations influenced by local and nearby sources and sinks, or long-range transport of emitted gases? (2) What are the different contributions (of anthropogenic, biogenic, biomass burning … fluxes) to the observed concentrations, both at the surface and in the total column? (3) How accurate is WRF-GHG in simulating the different observation types of the three gases ($CO_2$, $CH_4$ and CO)

in the Southern Indian Ocean region, in particular at Saint-Denis and at the Maïdo Observatory? What are its strengths and weaknesses?

The structure of this document is as follows. Section 2 describes the location of the observation sites at Reunion Island, the general transport patterns, the GHG-measuring instruments and the data sets used in this study. Details on the model set-up and input inventories are described in Sect. 3. Section 4 constitutes the main part of this manuscript. First, the model performance

is evaluated with regard to meteorological fields, both at the surface and higher up, in subsection 4.1. The model assessment





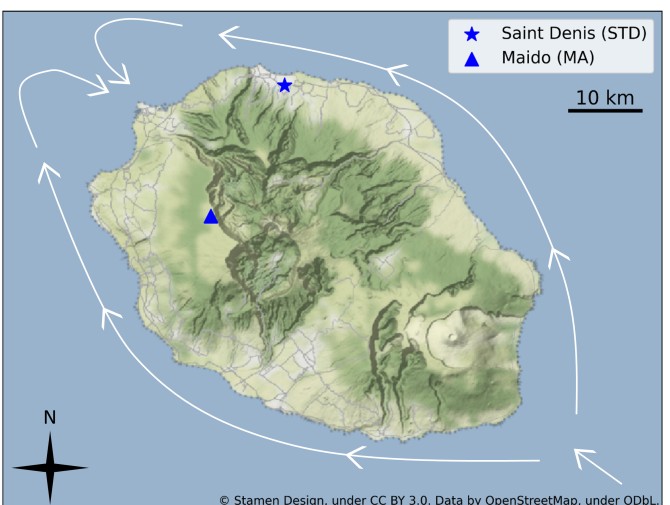

**Figure 1.** Map of Reunion Island, indicating the location of the two measurement sites: Saint-Denis (star) and Maïdo (triangle). The white arrows roughly illustrate the local wind patterns, generated by the trade winds and the orography of the island.

and data analysis at Saint-Denis and Maïdo are discussed in subsection 4.2 and 4.3 respectively. Finally, the impact of model resolution is discussed in Sect. 5 and conclusions are drawn in Sect. 6.

## 2   Observations at Reunion Island

The data used in this study are coming from two observation sites on Reunion Island: Saint-Denis (referred to as STD from now on; 20.9014 ° S, 55.4848 ° E; 85 m above sea level) which is the capital city and is situated close to the northern coast, and the Maïdo Observatory (referred to as MA from now on; 21.0796 ° S, 55.3841 ° E; 2155 m a.s.l.) which is close to the top of a mountain ridge on the northwest side of the island. Currently, each site is equipped with a Fourier Transform InfraRed (FTIR) spectroscopy instrument and an in situ cavity ring-down spectroscopy (CRDS) analyzer, which are described more in detail below. These instruments measure the column-averaged dry-air mole fractions and local near-surface mole fractions, respectively. The locations of both sites on the island are shown in Fig. 1.

### 2.1   Climate and transport patterns

The atmospheric transport around Reunion Island is controlled by the position of the Intertropical convergence zone (ITCZ) and the south Hadley cell (Baldy et al., 1996; Foucart et al., 2018). During a large part of the year, a strong subtropical high induces steady southeasterly trade winds near the surface and westerlies aloft. Hence, the air above Reunion Island is characterized by a wind (and temperature) inversion causing generally clear skies which are common during the dry season (Baldy et al., 1996; Lesouëf et al., 2011; Baray et al., 2013). Typically, this (colder) dry season lasts from May to November (Foucart et al., 2018).





In Austral summer (January to March) the ICTZ moves South, sometimes reaching Reunion Island. This results in weaker trade winds and often heavy rains, resulting in the (warmer) wet season in those months (Baldy et al., 1996; Foucart et al., 2018).

With its high altitudes (up to 3 000 m above sea level), Reunion Island represents a sudden obstacle for the stable southeasterly trade winds. In combination with the inversion layer, this causes a blocking on the windward side and wind flow splitting (and accelerating) around the island, to form counter-flowing vortices on the northwestern (lee) side (Lesouëf et al., 2011). This is illustrated by the white arrows in Fig. 1. Moreover, the split flow is under influence of thermally driven circulations (so-called trade breathing): nighttime downslope and land-breezes push the trade wind offshore whereas daytime upslope and sea-breezes allow the wind to pass over coastal areas (Lesouëf et al., 2011; Lesouëf et al., 2013). These circulations are the dominant (daily) wind pattern on the northwestern side of Reunion Island (where MA is situated) which is sheltered of the trade winds (Lesouëf et al., 2011; Baray et al., 2013; Guilpart et al., 2017; Verreyken et al., 2021).

## 2.2 Saint-Denis

Saint-Denis is the capital of Reunion Island, located by the coast in the Northern part of the island. As of 2018, there were 309 635 inhabitants in the metropolitan area of Saint-Denis, with a population density of about 1 100 per square kilometer. The city lies on a slope between the ocean and the nature reserve of La Roche Écrite (ultimately reaching a top of 2 276 m).

The observations at STD are made on top of a building at the University of Reunion Island (85 m above sea level). In situ mole fractions of $CO_2$ and $CH_4$ are measured by a CRDS analyzer (Picarro G1301) since August 2010, in collaboration with the Laboratoire de l'Atmosphère et des Cyclones (LACy), the Observatoire des Sciences de l'Univers de la Réunion (OSU-R) and the Laboratoire des Sciences du Climat et de l'Environnement (LSCE). The measurements are available with a time frequency of 1 min and the uncertainty on the measured mole fractions are about 0.1 ppm and 2 ppb, for $CO_2$ and $CH_4$ respectively.

Besides the surface in situ measurements, ground-based FTIR observations are also performed, providing mole fractions in an atmospheric column along the solar path. In September 2011, BIRA-IASB installed a high-resolution Bruker IFS 125HR FTIR at STD. This instrument is primarily dedicated to measuring the near-infrared (NIR: 4000 - 16000 cm$^{-1}$) spectra and contributes to TCCON (Wunch et al., 2011). The solar spectra are used to retrieve the total column-averaged dry-air mole fraction of $CO_2$, $CH_4$ and CO (De Maziere et al., 2017). The standard TCCON retrieval algorithm, called GGG2014, applies a profile scaling, therefore deriving information on the total column only and not on the vertical profile. TCCON measurements have been calibrated to WMO standards, so it is assumed that there are no systematic biases compared to in situ measurements (Wunch et al., 2010). More detail on both instruments can be found in Zhou et al. (2018).

## 2.3 Maïdo

The Maïdo Observatory (2155 m a.s.l.) is located close to the summit of a mountain with the same name which has an altitude of about 2 200 m.a.s.l. and is situated in the western part of the island. The Observatory is devoted to long-term atmospheric monitoring in the tropical region of the Southern Hemisphere and houses a variety of atmospheric measurement instruments such as lidar systems, spectro-radiometers, and in situ gas and aerosol analyzers (Baray et al., 2013). To the west of MA is a





gentle slope reaching the coastal areas and the ocean, while the summit lies to the east of the site, followed by a cliff leading to the caldera of Cirque de Mafate. The area around MA is covered by mountain shrubs and heathlands (Duflot et al., 2019).

The mole fractions of all three gases ($CO_2$, $CH_4$ and CO) have been collected by a CRDS analyzer (Picarro G2401) at MA since December 2014 and have been certified as Integrated Carbon Observation System (ICOS) atmospheric data in late 2019

(De Mazière et al., 2021). The measurements are available at a time resolution of 1 min and the uncertainties are about 50 ppb, 1 ppb and 2 ppb for $CO_2$, $CH_4$ and CO, respectively.

In March 2013, BIRA-IASB started operating a second Bruker IFS 125HR FTIR spectrometer, besides the one at STD, but observing the solar spectra in the mid-infrared (MIR) range from 600 to 4500 $cm^{-1}$ (Baray et al., 2013). These FTIR measurements are affiliated with NDACC. Gas mole fractions of $CH_4$ and CO are retrieved from the FTIR solar spectra by the SFIT4

algorithm which is based on the optimal estimation method of Rodgers (2000). More information about the specific methods used can be found in Zhou et al. (2018). The final data consist of the retrieved vertical profiles, expressed as volume mixing ratio (VMR) profiles on a vertical altitude grid.

## 2.4  Meteorological measurements

The quality of the WRF-GHG simulations is evaluated against meteorological fields that are being measured in parallel at the

two observation sites. More specifically, there are in situ measurements of 2 m temperature, 10 m wind direction and wind speed. These fields are measured by the Vaisala Weather Transmitter (model WXT510 at STD and model WXT520 at MA) every 3 s.

Additionally we will compare the WRF-GHG output with vertical profiles from operational daily meteorological Meteomodem M10 radiosonde launches performed by Météo-France at 12:00UTC at the Gillot airport (4 km away from STD). The

Meteomodem M10 radiosondes provide measurements of temperature, pressure and relative humidity with respect to water and zonal and meridional winds. Detailed description of this sensor can be found in Dupont et al. (2020).

## 3  WRF-GHG model

WRF-GHG is an abbreviation for the Weather Research and Forecast model coupled with Chemistry (WRF-Chem) in its passive tracer option (Skamarock et al., 2008; Beck et al., 2011). WRF-Chem simulates the emission, transport, mixing,

and chemical transformation of trace gases and aerosols simultaneously with the meteorology. In WRF-GHG only $CO_2$, CO and $CH_4$ are transported and there are no chemical reactions simulated. Separate tracers for each compound represent the contribution from the fluxes within the model domains (d01-d03) from different categories: anthropogenic, biomass burning, biogenic (for $CO_2$ and $CH_4$ (termites)), ocean (for $CO_2$ only) and wetlands (for $CH_4$ only). Additionally, there is a so-called background tracer which represents the contribution of the initial and lateral boundary conditions. The sum of all tracers for a

species is equal to the total modeled mole fractions. In this study, WRF-Chem version 4.1.5 is used.

Two time periods have been simulated: from 1 August 2015 until 1 May 2016 and from 1 July 2016 until 15 July 2017. These periods have been selected because then, quite complete datasets are available from all considered instruments. The first 14





days in each period are regarded as spin-up period and are not used in the model-data comparisons. The model provides 3D fields of $CO_2$, $CH_4$, $CO$ and meteorological fields every hour.

## 3.1 Emissions, initial and boundary conditions

An overview of the data that are used as input to the WRF-GHG model are given in Table 1 and described hereafter. The hourly meteorological initial and lateral boundary conditions (IC-BCs) are obtained from the European Centre for Medium-Range Weather Forecasts (ECMWF) global ERA5 reanalysis dataset (0.25 ° x 0.25 °) (Hersbach et al., 2018a, b), while the chemical IC-BCs are imported from the CAMS global reanalysis for greenhouse gases (EGG4, for $CO_2$ and $CH_4$) and reactive gases (EAC4, for $CO$, Inness et al. (2019)). The data for $CO_2$ and $CH_4$ are available every 3 h, while for $CO$ every 6 h. These fields from the CAMS reanalysis are used to drive the background tracers. The IC-BCs of the tracers corresponding with the contribution from surface fluxes are set to zero.

The anthropogenic emissions for $CH_4$ and $CO$ are taken from the Emission Database for Global Atmospheric Research (EDGAR): v5.0 Global Greenhouse Emissions product (Crippa et al., 2019b) for $CH_4$ and v5.0 Global Air Pollutant Emissions product (Crippa et al., 2019a) for $CO$. Further, we performed simulations over a short period of a couple days to test alternative inventories for anthropogenic $CO_2$ and $CO$ fluxes. We concluded that the Open-Data Inventory for Anthropogenic Carbon dioxide (ODIAC2020, Oda and Maksyuto (2015, 2011); Oda et al. (2018)) was more representative for the anthropogenic $CO_2$ emissions, probably due to its much higher spatial resolution (1 km) compared to EDGAR (0.1 °).

Similarly, we use a $CO$ surface emission inventory at a resolution of 500 m, based on the posterior estimates of a mesoscale inverse model (Jérôme Brioude, personal communication), but only in the innermost domain d03. The atmospheric transport of the inverse model was calculated using the FLEXPART Lagrangian dispersion model (Verreyken et al., 2019) coupled with the MESO-NH mesoscale model (Lac et al., 2018) at a resolution of 500 m and 60 vertical levels. FLEXPART-MESO-NH was run backward in time to calculate the source-receptor relationships between MA and the surface sources from the $CO$ measurements at MA from April 4th to May 3rd 2019, during the BIO-MAIDO campaign (Dominutti et al., 2022). The ODIAC $CO_2$ emission inventory was used as a priori to benefit from its native spatial resolution of major urban areas. A scaling factor, based on the ratio between the mean $CO$ enhancement above background and mean $CO_2$ enhancement above background, was applied on the $CO_2$ fluxes to obtain a priori surface $CO$ fluxes. A temporal resolution of one hour was used for the observed and simulated $CO$ mixing ratios at MA. A lognormal distribution was assumed for the observation and surface flux errors (Brioude et al., 2012, 2013). Such assumption better matches the $CO$ distribution in the atmosphere, and prevent the inversion to calculate negative fluxes.

The anthropogenic fluxes used within WRF-GHG are combined with a temporal emission factor from Nassar et al. (2013). Note that these factors are representative for $CO_2$ and might be less accurate for $CO$ and $CH_4$.

Daily biomass burning emissions for all three gases are obtained from the Fire INventory from NCAR (FINN v1.5) (Wiedinmyer et al., 2011). The biogenic $CH_4$ flux from wetlands is obtained from the WetCHARTs v1.0 ensemble (Bloom et al., 2017), while the biogenic $CO_2$ flux from oceans is taken from the observation-based global monthly gridded sea surface $pCO_2$ climatology by Landschützer et al. (2017), which also provides air-sea $CO_2$ fluxes. Finally, the biogenic $CO_2$ flux from the





vegetation is simulated online using the VPRM model (Mahadevan et al., 2008; Ahmadov et al., 2007). This model uses the 2 m temperature and downward shortwave radiation calculated by WRF-GHG in combination with surface reflectance data from the Moderate Resolution Imaging Spectroradiometer (MODIS). Further, it uses the global SYNMAP land cover data of 210 1 km resolution by Jung et al. (2006). Additionally, the VPRM requires a set of four model parameters for each vegetation class, dependent on the region of interest. Ideally, these parameters are optimized using a network of eddy flux towers. Since this is not available at Reunion Island, we use the set of parameters optimized by Botía et al. (2021), based on measurements from nine sites in the Amazon region in Brazil, created in the context of the Large Scale Biosphere-Atmosphere Experiment (LBA-ECO). Exact parameter values are given in Table A1 of appendix A.

| | Species | Source | Time and spatial resolution |
|---|---|---|---|
| Initial and lateral boundary conditions | $CO_2$, $CH_4$ | CAMS reanalysis for greenhouse gases | 3-hourly, 0.75 ° |
| | CO | CAMS reanalysis for reactive gases | 6-hourly, 0.75 ° |
| Anthropogenic flux (multiplied with temporal factors of Nassar et al. (2013)) | $CO_2$ | ODIAC2020 | Monthly, 1 km (land), 1 °(ocean) |
| | $CH_4$ | EDGAR v5.0 Global Greenhouse Emissions | Yearly, 0.1 ° |
| | CO | EDGAR v5.0 Global Air Pollutant Emissions (d01, d02) | Yearly, 0.1 ° |
| | | Brioude (d03) | Yearly, 500 m |
| Biomass burning flux | $CO_2$, $CH_4$, CO | FINN v1.5 | Daily, 1 km |
| Biogenic flux | $CO_2$ | online (VPRM) | Hourly, model resolution |
| | $CH_4$ | online, WetCHARTs v1.0 | Monthly, 0.5 ° |
| Ocean flux | $CO_2$ | Observation-based global monthly gridded sea surface $pCO_2$ climatology | Monthly, 1 ° |

**Table 1.** Overview of data sets used as input for the WRF-GHG simulations.

## 3.2 Settings

To achieve a high-resolution model grid over Reunion Island, a configuration of three nested domains was established, going from a larger domain with lower resolution to a smaller domain with higher resolution. The domains are shown in Fig. 2. Their respective resolutions are 50 km, 10 km and 2 km. The innermost domain d03 covers Reunion Island and the two measurement sites completely. WRF-GHG uses a hybrid vertical coordinate, which is a coordinate that is terrain following near the ground 220 and becomes isobaric higher up. In all our domains the model has 60 vertical levels extending from the surface up to 50 hPa. The following physical parameterization options are used: the Morrison 2–moment Scheme (Morrison et al., 2009) for micro physics, the RRTMG Shortwave and Longwave Schemes (Iacono et al., 2008) for shortwave and longwave radiation. The Eta Similarity Scheme (Janjić, 1994) for surface layer processes and the Unified Noah Land Surface Model (Tewari et al., 2004) for the land surface. To choose between the diverse parameterization schemes for cumulus parameterization and planetary bound- 225 ary layer (PBL) physics, several model test runs were made for a short simulation period of a couple of days and compared





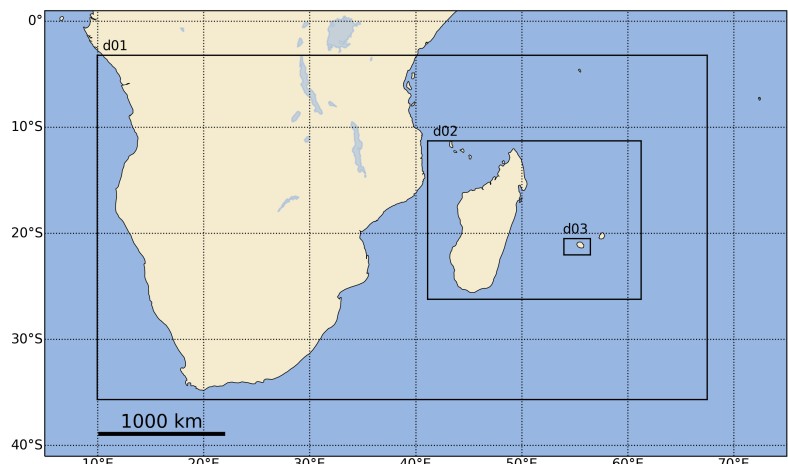

**Figure 2.** Location of the WRF-GHG domains, with horizontal resolutions of 50 km (d01), 10 km (d02) and 2 km (d03).

with the observed meteorology. As a result, the University of Washington (TKE) Boundary Layer Scheme (Bretherton and Park, 2009) for PBL physics and the Grell–Freitas Ensemble Scheme (Grell and Freitas, 2014) for cumulus parameterization, but only in the largest domain (d01), were chosen for this study.

## 3.3 Data handling

The various observation types are dealt with in different ways for comparison with the model.

The surface observations (both meteorological fields and GHGs) are averaged over a period of 30 min around the hourly model time step. At STD, we compare these data with the lowermost level of the model grid cell whose center is closest to the location of the instrument. Because of the complex topography, the cell covering MA is less representative for the Observatory as its center is located behind the summit, below in the caldera of Cirque de Mafate. Model-data comparisons of the surrounding

cells showed that the cell to the West of MA is more representative. Therefore, this alternative model grid cell, of which the center is only 1.3 km away from the Observatory, is used in the analysis.

In order to compare similar quantities, the total-column-averaged dry-air mole fractions from TCCON and NDACC are truncated to the same atmospheric column that is simulated by WRF-GHG, e.g. from surface up to 50 hPa. This is needed because the FTIR data represent the total atmospheric column whereas the WRF-GHG upper limit lies at around 21 km.

As NDACC additionally provides volume mixing ratio profiles, the column-averaged mole fractions are re-calculated taking only those layers below the model upper limit. For TCCON, only information on the total column is retrieved. Therefore, we multiply the TCCON data with a factor representing the ratio between the column-averaged mole fraction of the smaller column (up to 50 hPa) to that of the total column. This ratio is calculated from the a priori information. In the rest of this manuscript, all dry-air column-averaged mole fractions (so-called Xgas) mentioned in this article refer to this reduced atmospheric column

only (surface up to 50 hPa). Due to the specific profile of the respective gases in the atmosphere, this scaling is more significant





for XCH$_4$, than it is for XCO or XCO$_2$: the values generally increase after scaling by about 27-35 ppb for XCH$_4$, 3 ppb for XCO and 0.25 ppm for XCO$_2$.

To compare with the hourly WRF-GHG outputs, the scaled mole fractions are averaged over a period of 30 min around the model time step. Furthermore, a smoothing is applied to the WRF-GHG profiles, according to Rodgers and Connor (2003).

Because of the different characteristics of the TCCON and NDACC observing systems, this smoothing procedure is slightly different at the two sites. Technical details on how the smoothed dry-air column-averaged mole fractions of WRF-GHG are calculated at STD and MA can be found in Appendix B.

## 4   Results

### 4.1   Meteorological evaluation

#### 4.1.1   Surface measurements

To assess the general model performance, the hourly model output of d03 near the surface is compared with local measurements of 2 m temperature and 10 m wind direction and speed at both sites. Table 2 gives the root mean square error (RMSE), mean bias error (MBE) and Pearson correlation coefficient (CORR) of the model-data comparison over the complete time series (13 583 paired data points at STD, 14 031 at MA).

|  | 2 m temperature (C) | | 10 m wind direction (°) | | 10 m wind speed (ms$^{-1}$) | |
| --- | --- | --- | --- | --- | --- | --- |
|  | STD | MA | STD | MA | STD | MA |
| RMSE | 1.33 | 1.94 | 52.33 | 66.80 | 4.29 | 2.93 |
| MBE | 0.74 | -0.35 | / | / | 3.83 | 1.59 |
| CORR | 0.93 | 0.83 | 0.72 | 0.76 | 0.73 | 0.27 |

**Table 2.** Overview of the meteorological evaluation of surface measurements at the two sites.

The 2 m temperature is well simulated by WRF-GHG at both sites with very high correlation coefficients of 0.93 at STD and 0.83 at MA, and RMSE between 1 ° and 2 ° Celsius (1.33 at STD, 1.94 at MA). Figure 3a-b compares the median diurnal cycle at both sites, which is very well reproduced by the model. Overall, higher temperatures are measured at STD compared to MA because of the large difference in altitude between the sites (85 masl compared to 2 155 masl).

     The windroses in Fig. 4 show the most occurring 10 m wind direction and their corresponding wind speed. The 10 m wind

direction of WRF-GHG correlates well with the measurements at both sites (correlation coefficients of 0.72 and 0.76). There is a larger error of the wind direction at MA (RMSE of 66.80 °) compared to STD (RMSE of 52.33 °). At STD, the wind is mainly from east or southeast (trade winds), however for calmer wind speeds (< 2 ms$^{-1}$) the wind can also come from the south(west). WRF-GHG captures the dominant southeastern winds, but does not simulate winds from the south. It highly overestimates the wind speed, with a mean bias error of 3.83 ms$^{-1}$ and a RMSE of 4.29 ms$^{-1}$. There is a clear diurnal cycle

of the wind speed at STD, shown in Fig. 3c, with stronger winds during the day and calmer conditions at night. As WRF-GHG

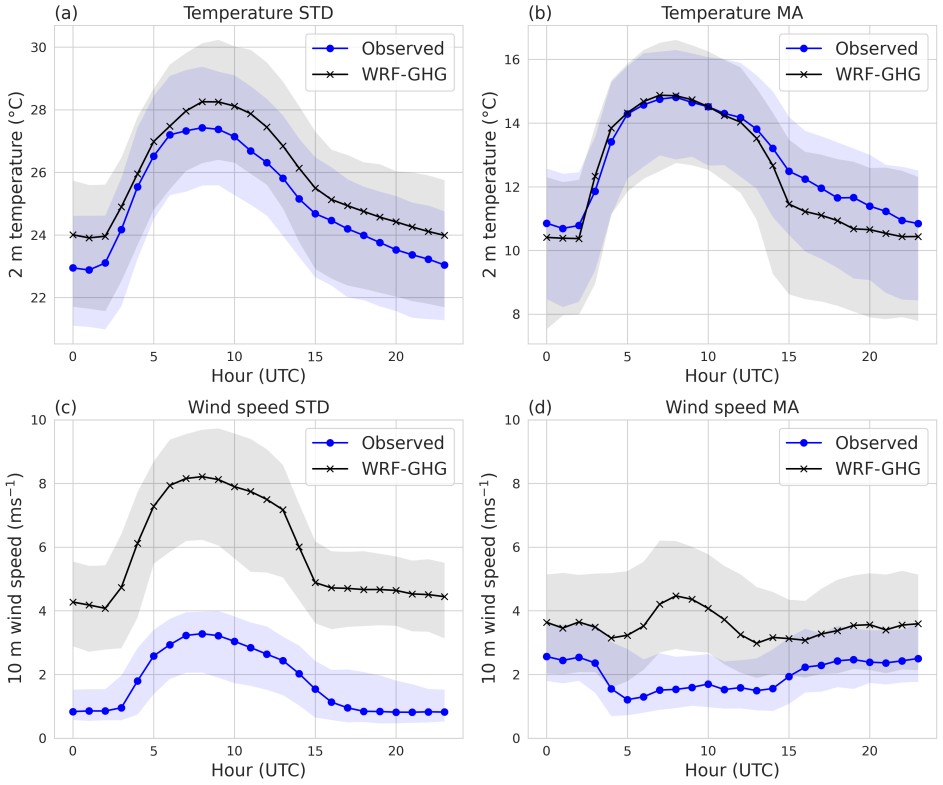

**Figure 3.** (a)-(b) Diurnal cycle of the 2 m temperature and (c)-(d) 10 m wind speed at both STD and MA. The blue and black lines show the median values for every hour of the measurements and simulations, respectively. The shaded blue and grey areas indicate the corresponding interquartile ranges of the measurements and simulations, respectively. Hours are given in UTC (local time at Reunion Island is UTC+4). Remark that the temperature plots have different y-axes.

follows the observed pattern, the correlation coefficient is still quite high (0.76). The overestimation might be caused by an underestimation of the surface roughness of the city within WRF-GHG. Besides the unified NOAH land surface model (see Sec. 3.2), no additional urban surface model was included in the simulations. Other studies using the WRF model often show wind speed overestimation above urban areas (Feng et al., 2016; Barlage et al., 2016; Zhang et al., 2009; Kim et al., 2013).

Additionally, there is a large gradient in the surface wind speed near STD, caused by the presence of the strong trade winds. Therefore, an insufficient high model resolution might also be the cause for the wind speed overestimation.

At MA on the other hand, the most common wind direction is east with some occurrences of west winds. This points to the typical thermally induced circulations during the day, whereby wind is driven from the coast upwards and sometimes reaches MA (Duflot et al., 2019). The prominent east winds illustrate the presence of overflowing trade winds. The simulated winds

from WRF-GHG are mainly from east, indicating that the larger errors at MA might be linked to the missing western wind components. This is likely due to the complex topography around the Maïdo Observatory and the model resolution (of 2 km) which might be insufficient to resolve these very local wind dynamics.



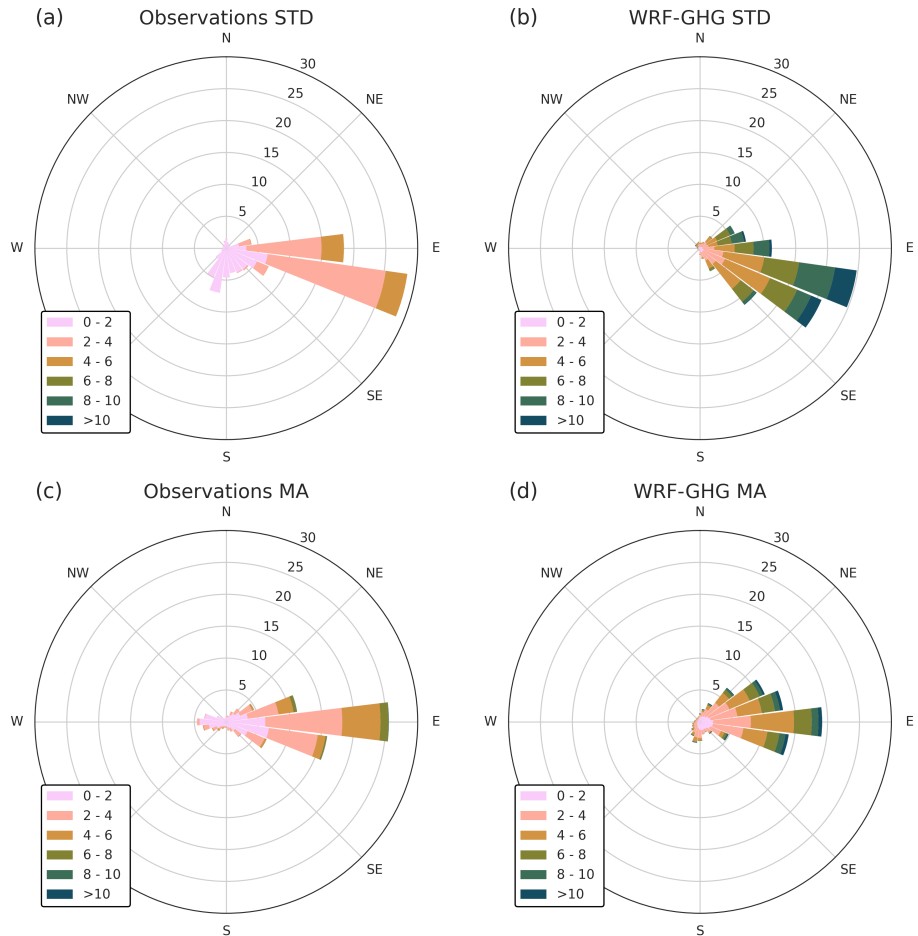

**Figure 4.** Windrose from observations and WRF-GHG simulations at STD ((a)-(b)) and at MA ((c)-(d)). The colors indicate the associated wind speed (in $\mathrm{ms^{-1}}$), while the lengths of the bars show the frequency of any wind direction binned by $15\,°$, given in percentage.

As to the wind speed at MA, the bias and RMSE are smaller (1.59 and 2.93 $\mathrm{ms^{-1}}$, respectively) than at STD but the model is still overestimating the wind speed. Moreover, the correlation is very low at this site. The daily 10 m wind speed cycle at MA

is less distinct than at STD, however at night the wind is more often faster ($> 2\,\mathrm{ms^{-1}}$) than during the day. This could be linked with the local wind dynamics around MA where during the day calmer upslope winds from the west often reach MA while at night the Observatory is generally in the free troposphere under influence of the faster trade winds (Guilpart et al., 2017).

### 4.1.2 Radiosonde profile measurements

Daily radiosonde profiles of air temperature, wind direction, wind speed and relative humidity are compared with the model

data to assess the accuracy of WRF-GHG on all levels of the troposphere. The profiles were matched as follows: for every data point measured by the radiosonde, the grid cell corresponding to its coordinate is selected. Next, the model profile (consisting



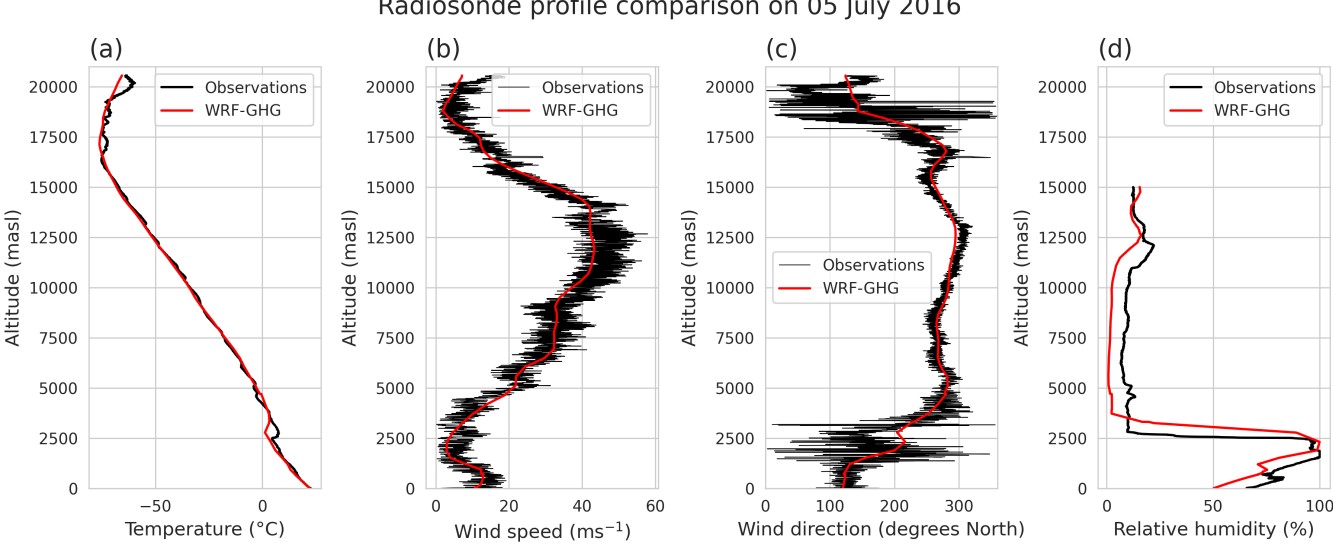

**Figure 5.** Example of the radiosonde data at Gillot airport on 5 July 2016, compared with the model for (a) temperature, (b) wind speed, (c) wind direction and (d) relative humidity. The black line represents the measured values. The red line is the corresponding WRF-GHG data.

of the meteorological field in the complete vertical column above the selected grid cell) is interpolated to the altitude of the measurement. This interpolated value is then paired with the value of the measurement. This results in a paired model-data profile for the four variables once every day. An example of such a paired profile on 5 July 2016 is shown in Fig. 5. For every 295 paired profile, the RMSE, MBE and CORR statistics are calculated. This is done for a total of 267 days in the year 2016.

For the temperature, correlation coefficients are very high on all days (median is $0.99$). Moreover, the RMSE values are quite small on most days (median is $1.07\,^{\circ}$C) indicating that WRF-GHG can simulate the temperature in the troposphere quite accurately.

There is a good correlation for the wind direction and speed profiles as well. Half of the days in 2016 have a correlation coefficient higher than $0.87$ for wind direction and $0.83$ for wind speed. When calculating the RMSE of wind direction along the daily profiles, we find a median of $48.18\,^{\circ}$ while the median RMSE of wind speed is $3.87\,\mathrm{ms}^{-1}$. On most days, WRF-GHG is slightly underestimating the wind speed (median bias error is $-1\,\mathrm{ms}^{-1}$), which is in contrast with the overestimation found at the surface sites.

The profiles of relative humidity are analysed up to an altitude of 15 km because the measurements are less accurate higher up. WRF-GHG correlates well with these profiles, with a correlation coefficient higher than $0.87$ on 75 % of the days in 2016. The median RMSE on the daily profiles of relative humidity is only 11.6 %, showing a decent model performance.

Overall we can conclude that the simulations of basic meteorological parameters are quite accurate along vertical profiles, where near the surface wind speed and direction agree less with the observations.





## 4.2 GHG data at Saint-Denis


At STD, the in situ surface mole fractions of $CO_2$ and $CH_4$ are measured together with the TCCON column-averaged dry-air
mole fractions of $CO_2$, $CH_4$ and CO. The comparison with the WRF-GHG simulations will be described in detail below, for
each species and measurement type separately. Full time series of the observed and modeled data can be found in appendix C.
An overview of the statistics of the comparisons is given in Table 3.

| | STD | | | MA | | |
|---|---|---|---|---|---|---|
| | (X)$CO_2$ | (X)$CH_4$ | (X)CO | (X)$CO_2$ | (X)$CH_4$ | (X)CO |
| | (ppm) | (ppb) | (ppb) | (ppm) | (ppb) | (ppb) |
| *in situ* | | | | | | |
| RMSE | 9.17 | 18.51 | / | 1.95 | 19.33 | 10.99 |
| MBE | -5.39 | 9.04 | / | -0.15 | 14.09 | 5.51 |
| CORR | 0.62 | 0.35 | / | 0.75 | 0.30 | 0.83 |
| *FTIR* | | | | | | |
| RMSE | 0.75 | 10.26 | 8.08 | / | 10.80 | 7.37 |
| MBE | -0.37 | 5.69 | 5.07 | / | -5.65 | 1.81 |
| CORR | 0.90 | 0.31 | 0.89 | / | 0.37 | 0.90 |

**Table 3.** Overview of the WRF-GHG performance for hourly in situ and column observations of GHG at Reunion Island. Comparison with
the column observations is based on the smoothed model profiles. There are no in situ CO data available at STD (Saint-Denis) and no $XCO_2$
data at MA (Maïdo).

### 4.2.1 Surface $CO_2$


The model-data comparison of the surface data shows a moderate correlation coefficient of $0.62$ together with a relative large
error of 9.17 ppm and a model underestimation of 5.39 ppm. The scatter plot in Fig. 6a indicates that these discrepancies arise
from a model underestimation of the higher $CO_2$ mole fractions. The lower $CO_2$ concentrations are in general much better
reproduced.

The $CO_2$ measurements at STD show a clear diurnal cycle (see Fig. 7a), with lower values during the day and higher values
during the night. The diurnal cycle of WRF-GHG reproduces this pattern but with much lower nighttime concentrations,
leading to the moderate correlation found in Table 3.

As shown in the diurnal cycle in Fig. 7b, the main contributors to the total $CO_2$ signal in WRF-GHG, in addition to the
background signal, are the anthropogenic and biogenic tracers. They correspond with anthropogenic and biogenic fluxes within
the model domains (d01-d03) and show similar diurnal patterns with maxima at night and minima during the day. The influence
of biomass burning or ocean fluxes are negligible at STD.

In urban areas, anthropogenic pollution is generally trapped in and around the city creating a so-called urban $CO_2$ dome (Idso

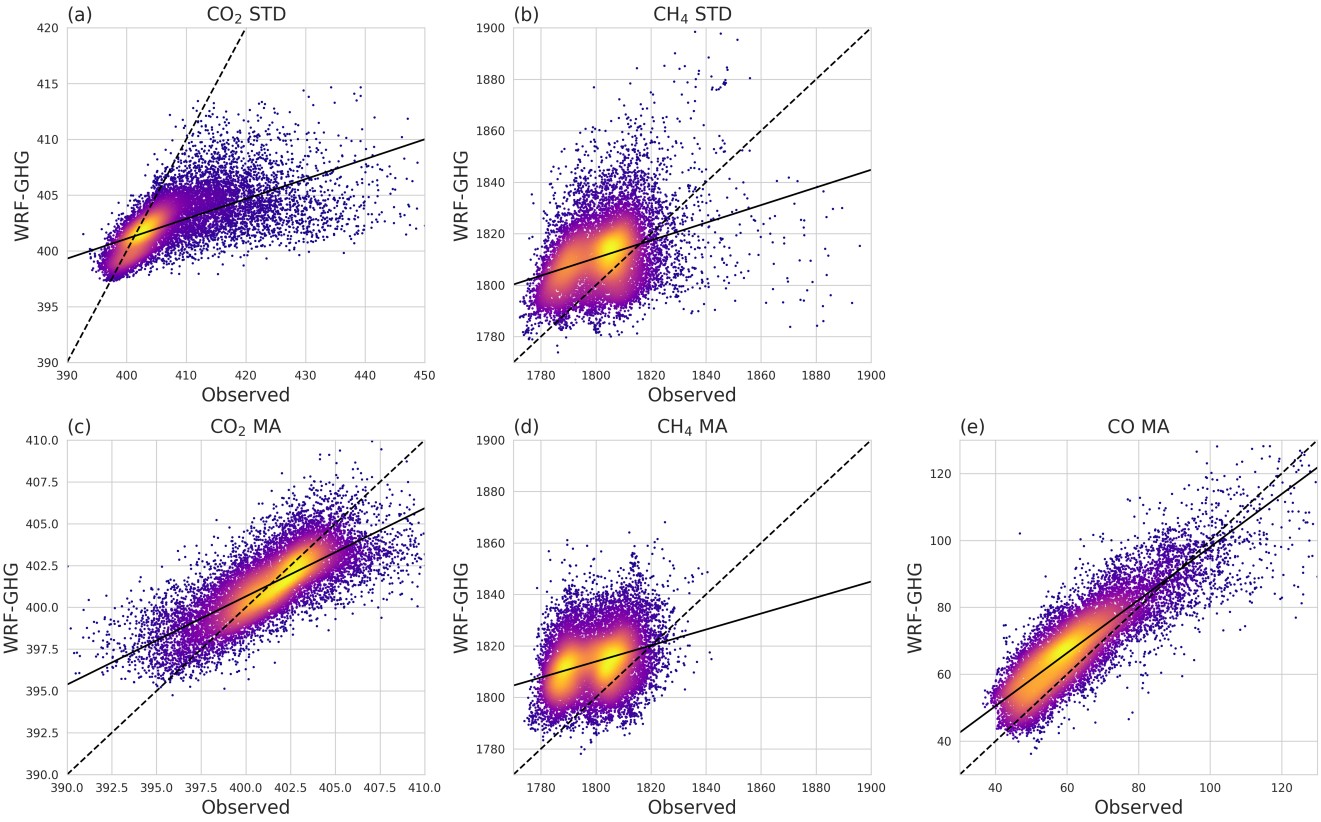

**Figure 6.** Scatter plot of hourly observed and modeled in situ greenhouse gases at STD (a,b) and MA (c,d,e). The colors indicate the point density.

et al., 2002). The strength of this dome is primarily dependent on the local emissions and variations in the boundary layer. In calm weather, near-surface air temperature inversions at night trap anthropogenic pollution near the ground in the shallow

nocturnal boundary layer, leading to strongly enhanced $CO_2$ mixing ratios. During the day, solar radiation causes convective mixing of the air creating a deep planetary boundary layer (PBL). The near-surface $CO_2$ concentrations are then diluted by this thorough mixing of air, and the urban dome extends to greater heights.

However, wind speed and direction can alter the strength of this urban $CO_2$ enhancement: at higher wind speeds (from $2\,\mathrm{ms}^{-1}$), ventilation processes prevent strong $CO_2$ accumulation, while winds from rural areas could bring pristine air to the city (Idso

et al., 2002; Rice and Bostrom, 2011; Massen and Beck, 2011; García et al., 2012; Xueref-Remy et al., 2018).

Within WRF-GHG, the main contributors to the simulated $CO_2$ in the grid cells around STD are anthropogenic and peak during the day. The biogenic $CO_2$ flux at the grid cell of STD is zero because it is assumed that there is no vegetation within the city. Given that Saint-Denis is the capital city of Reunion Island and has plenty of anthropogenic activities, these WRF-GHG fluxes appear realistic. The nighttime peak of $CO_2$ mixing ratios is therefore attributed to PBL dynamics and regional transport.

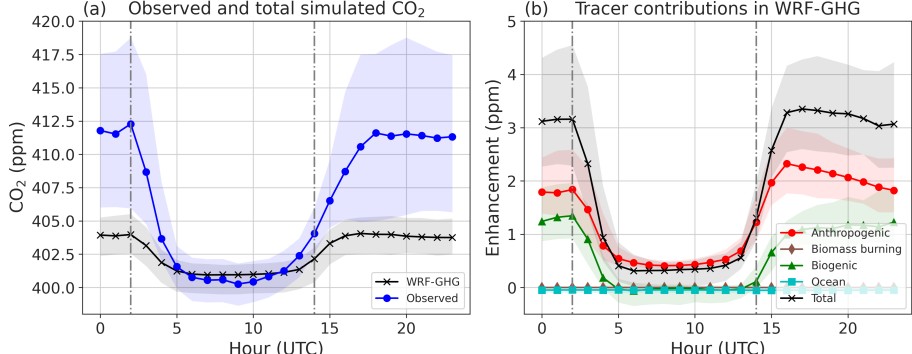

**Figure 7.** Diurnal cycle of (a) in situ $CO_2$ at STD (b) and model tracer contributions. The black line in (a) represents the median hourly concentrations of WRF-GHG, while the blue line represents the observed values. The shaded areas cover the interquartile ranges. The gray dotted vertical lines at 2 h and 14 h UTC indicate the approximate times of sunrise and sunset. The colored lines in (b) represent the different tracer deviations from the background concentration in WRF-GHG: anthropogenic (red), biogenic (green), ocean (cyan) and biomass burning (brown) tracer. The black line is the sum of all tracers, except the background.

Figure 7a shows that the observed interquartile range at night is wide, indicating a large variability in the $CO_2$ accumulation. We find a negative correlation between 10 m wind speed and in situ $CO_2$ concentrations at night for the observations at STD (see Fig. 8). At low wind speeds, a large variability in $CO_2$ mixing ratios is observed: from 400 ppm up to 450 ppm. For wind speeds above 2 ms$^{-1}$ on the other hand, values higher than 410 ppm are rarely found, which indicates that ventilation processes take place. At the same time, there is a large overestimation of the wind speed within WRF-GHG (Sec. 4.1.1, Table 2). At night,
91.7 % of the simulated hours has a wind speed of more than 2 ms$^{-1}$, compared to only 23.2 % of the nocturnal observations, leading to a wind speed MBE of 3.23 ms$^{-1}$ at night. Note that the mean wind speed at night within WRF-GHG is 4.4 ms$^{-1}$ (see also Fig. 3c), while the nighttime $CO_2$ concentration in WRF-GHG is on average 403.8 ppm (see also Fig. 7a). Looking at Fig. 8, these values follow the pattern as found in the observations: a nocturnal wind speed of more than 4 ms$^{-1}$ corresponds with $CO_2$ mole fractions of about 403-404 ppm. Therefore the model is likely underestimating the in situ $CO_2$ observations at
STD because of an overestimation of the surface wind speed.

 Further, we examine the relation of nighttime $CO_2$ concentrations and 10 m wind direction at STD. Figure 9a shows that the dominant observed wind direction at night is east-southeast (ESE), followed by south (S). The ESE winds generally correspond with higher wind speeds (> 2 ms$^{-1}$) and lower $CO_2$ concentrations (generally below 410 ppm), whereas observations with south winds generally coincide with very low wind speeds and $CO_2$ accumulation (Fig. 9a and Fig. 9b). The region in the
east-southeast of STD is a rural area dominated by agricultural activities. Therefore these stronger ESE winds would generally bring lower-$CO_2$-content air to STD. WRF-GHG, on the other hand, overestimates the wind speed and almost consistently simulates ESE winds and lower $CO_2$ concentrations (Fig. 9c and Fig. 9d). As such, the model-data mismatch is likely caused





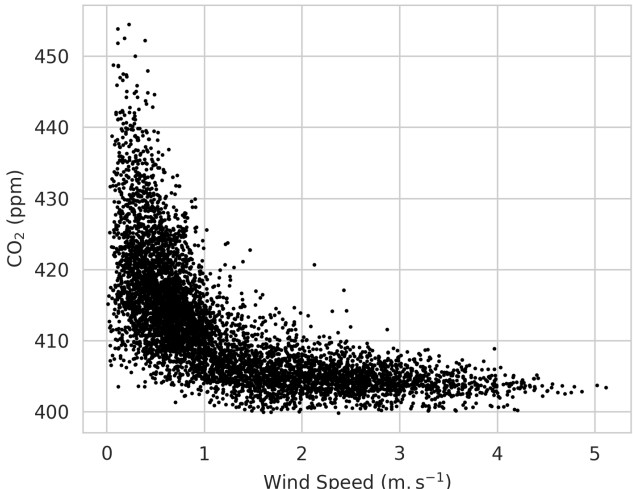

**Figure 8.** Scatter plot of nighttime wind speed at STD against hourly observed in situ $CO_2$. Nighttime hours are defined as those between 14 UTC and 2 UTC.

by a combination of both wind speed overestimation in WRF-GHG and discrepancies as to the wind direction, which are interrelated.

### 4.2.2   XCO₂

A higher correlation coefficient of $0.90$ is found when comparing the daytime hourly-averaged TCCON $XCO_2$ data with the smoothed $XCO_2$ from WRF-GHG, see Table 3. Moreover, the RMSE along the time series is $0.75$ ppm and there is a model underestimation of $0.37$ ppm. The hourly relative model-data errors are below $0.5$ % (Fig. 10). This is however slightly larger than the TCCON standard deviation of the hourly averages, which is around $0.1$ %.

WRF-GHG provides a separation into different tracer contributions, which are shown in Fig. 11. Monthly averages of these contributions for $XCO_2$ show a large (positive) biogenic enhancement in the months August to December. The biogenic tracer in WRF-GHG is driven by the online biogenic $CO_2$ fluxes calculated through the VPRM module as the sum of the gross ecosystem exchange (GEE) and respiration (Mahadevan et al., 2008). A positive biogenic tracer suggests that the nighttime respiration accumulates more $CO_2$ than the ecosystem can capture during the day by photosynthesis. Indeed, in the Southern

Hemisphere, the dry season is generally from May until November leading to a decrease in GEE in some ecosystems (Quansah et al., 2015; Räsänen et al., 2017). Moreover, this carbon source was higher in 201 because of a strong El Niño event leading to higher temperatures and less precipitation in the Tropics (Yue et al., 2017).

The anthropogenic enhancement is relatively constant throughout the year. There is also a small biomass burning component modeled in $XCO_2$ in the months August to December, which corresponds to the biomass burning (BB) season. During these

months, frequent fires occur in southern Africa and America. Duflot et al. (2010) showed that these polluted air masses can



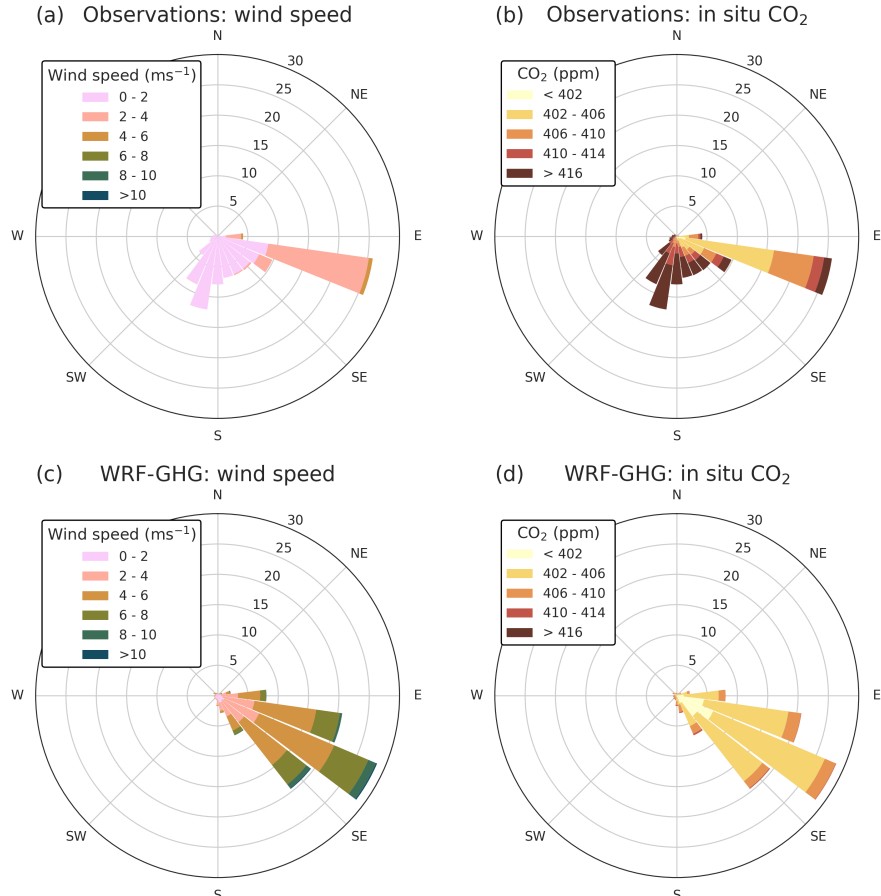

**Figure 9.** Wind rose of hourly nighttime data at STD, where the night is defined between 14 UTC and 2 UTC. (a) and (b) show the distribution of the observed wind speed per wind direction and near-surface $CO_2$ concentration per wind direction, respectively. (c) and (d) show the same for WRF-GHG simulated data. The lengths of the bars show the frequency of occurrence in percentage.

be transported to Reunion Island and detected by FTIR observations such as XCO. Note that the $XCO_2$ enhancements due to biomass burning coincides with the biogenic enhancements because, especially in the tropics, the occurrence and the duration of the BB season are linked to the dry season (Giglio et al., 2006).

As expected, the column observations of $CO_2$ are determined by different processes than the in situ $CO_2$ concentrations. Where the variation of $XCO_2$ is mainly driven by fluxes on the African continent, the surface $CO_2$ mole fractions are heavily influenced by local sources and PBL dynamics. This also agrees with the trajectory calculations by Zhou et al. (2018), showing that surface air mainly originates in the Indian Ocean while free tropospheric air is mainly coming from Africa and South America.





**Figure 10.** Relative percentage differences between hourly (smoothed) WRF-GHG and FTIR observations of $XCO_2$, $XCO$ and $XCH_4$. The blue dots represent the data at STD (WRF - TCCON), while the orange data are from MA (WRF - NDACC).

**Figure 11.** Monthly mean tracer contributions to the column-averaged mole fractions of (a) $CO_2$, (b) CO and (c) $CH_4$. The different colours represent different tracers: anthropogenic (red), biogenic (green), ocean (blue), wetlands (dark green), termites (light green) or biomass burning (brown). The solid lines are the mean monthly contributions at STD, while the dashed lines are for MA.





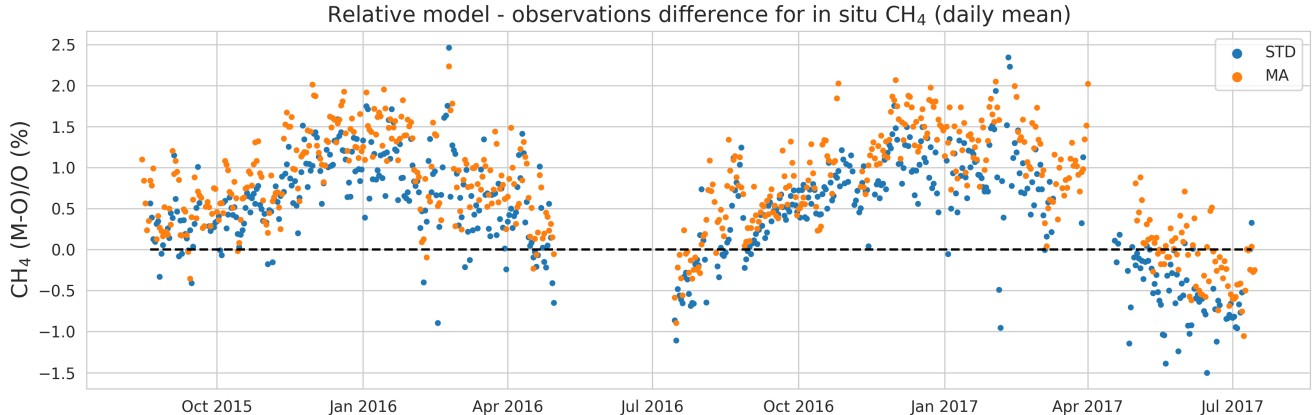

**Figure 12.** Time series of daily mean relative percentage differences between model and in situ observations of CH$_4$ at STD (orange) and MA (blue). The black dashed line indicates zero difference.

### 4.2.3 Surface CH$_4$

The model-data comparison for CH$_4$ at STD (Table 3) shows only a weak correlation (0.35) between modeled CH$_4$ and observations at STD. WRF-GHG shows an overestimation of about 9 ppb. The errors are however not constant over time: in Fig. 12 a seasonal bias is found, with larger errors between December and February, which is Austral summer. This is a known weakness in the CAMS reanalysis, used as boundary information, as pointed out in the most recent validation report by Ramonet et al. (2020): "For all observations CH$_4$ shows a seasonality in the relative difference between observations and

CAMS simulations, which is increasing in the Southern Hemisphere after 2008. ... The seasonal dependence, which needs to be investigated in more details, may be related to the representation of OH in the model, or/and to errors in the seasonal cycle of surface emissions (mainly from agriculture and wetlands)." This demonstrates the importance of accurate lateral boundary conditions to simulate long-lived tracers with regional models such as WRF-GHG.

The diurnal cycle of the CH$_4$ tracer contributions in Fig. 13 shows that the modeled CH$_4$ consists almost entirely of the

background signal and an anthropogenic enhancement, whereby both factors can add to the model-data mismatch.

The diurnal cycle is less pronounced in the observations (not shown). Nighttime values are on average only slightly larger than during the day, with a mean difference of only 3.09 ppb ($\sigma^2$=6.35), indicating that a nocturnal accumulation as identified for CO$_2$ in Saint-Denis is less evident for CH$_4$. Moreover, the overestimation of the daily amplitude in WRF-GHG suggests an overestimation of the local anthropogenic CH$_4$ fluxes from EDGAR.

In contrast to CO$_2$, the CH$_4$ mole fractions near the surface are less impacted by PBL dynamics and more by the background concentration.



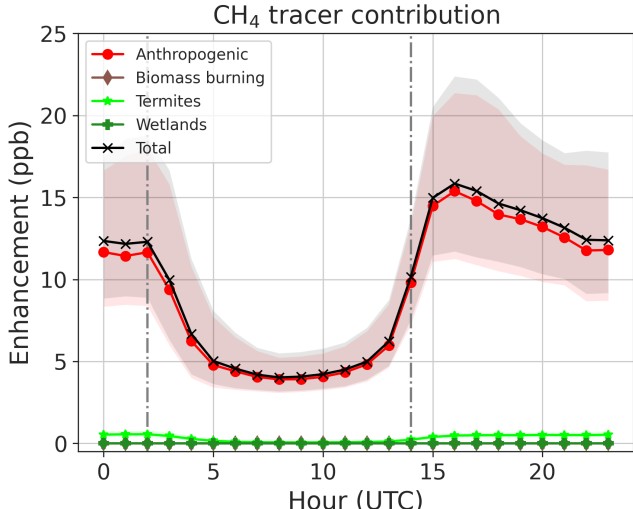

**Figure 13.** Diurnal cycle of in situ $CH_4$ tracer contributions of WRF-GHG at STD. The black crosses represent the median hourly enhancements above background for the sum of all tracers. The separate tracer contributions are given in red (anthropogenic), dark green (wetlands), light green (termites) or brown (biomass burning). The shaded areas cover the interquartile ranges. The gray dotted vertical lines at 2 h and 14 h UTC indicate the approximate times of sunrise and sunset.

### 4.2.4 $XCH_4$

Incorrect boundary values and hence background concentrations have an impact on all $CH_4$ simulations at Reunion Island. Therefore the statistics for the column-averaged mole fraction of $XCH_4$ at STD (part of TCCON) are worse than for $CO_2$. A
weak correlation is found (0.31, Table 3) and the model overestimates TCCON $XCH_4$ by 5.69 ppb, which is slightly less than the bias with respect to in situ data (Sec. 4.2.3).

Figure 10c shows the relative differences between WRF-GHG and observational data, which have the same seasonal pattern as for the in situ comparisons (Fig. 12) caused by the reported seasonal bias for $CH_4$ in the CAMS reanalysis data. The relative differences are below 2 %, but due to this seasonality in the errors, very little correlation is found.
Even though the model fails at reproducing the measured time series, it is still interesting to examine the different modelled tracer contributions to $XCH_4$ (Fig. 11b). The tracers contribute only a couple of ppb to the total signal, with the anthropogenic being dominant throughout the year. Further, small peaks in biomass burning enhancements are found during the BB season as for the other species. The biogenic tracers for $CH_4$ in WRF-GHG are generated by emissions from termites and wetlands. The termite signal is however very small and thus not relevant for this region. The signal from wetlands is larger, especially in
Austral summer. This roughly coincides with the rain season, causing greater wetland extent (Lunt et al., 2019).

In the same way as for $CO_2$, the surface $CH_4$ mole fractions at STD are influenced by local sources at Reunion Island, while fluxes from Africa and Madagascar are detected in the column observations because of the different air masses they sample.





### 4.2.5 XCO

At STD, CO is only available as a column-averaged mole fraction (part of TCCON). As seen in Table 3, a very high correlation
(0.89) is found for the hourly averaged paired data. WRF-GHG slightly overestimates the observed XCO (MBE: 5.07 ppb).
Figure 10c shows that the relative error between WRF-GHG and the XCO observations from TCCON is often below 20 % but
not constant: larger errors up to 30 % are found from January until May.

As for the other species, large contributions of biomass burning (BB) emissions are found in the months August to December
(Fig. 11c). Duflot et al. (2010) showed that XCO values during the BB season can reach up to twice the CO background
concentration from other months. The rather limited BB enhancement found in WRF-GHG suggests that a substantial amount
of the XCO increase in the BB season is already included in the background tracer. This suggests that fires outside of the
large domain can also be detected at Reunion Island, such as those from South America, which would confirm Duflot et al.
(2010). The anthropogenic contribution is more constant throughout the year and the dominant contribution outside BB season.
However it remains rather small compared to background, which appears to be the main driver behind the simulated XCO
values at Reunion Island.

The larger model overestimation in January 2016 and April 2017 are thus likely linked to the background tracer, which is based
on the CAMS global reanalysis for reactive gases. The corresponding CAMS validation report (Errera et al., 2021) mentions
no known biases, but shows similar relative errors in those months in the Southern Hemisphere and in particular at MA (visible
in Figure S.6 on p10). Again this demonstrates the importance of accurate boundary conditions for simulating XCO in this
region, but additionally points to the large influence of remote regions (outside domain d01) on the observed XCO time series
at Reunion Island.

### 4.3 GHG data at Maïdo

At MA, the surface mole fractions of all three gases ($CO_2$, $CH_4$ and CO) are measured together with the column-averaged
mole fractions of $CH_4$ and CO that are part of NDACC. The results of each species are given in the sections below. Again, full
time series of the observed and modeled data can be found in appendix C.

### 4.3.1 Surface $CO_2$

At MA, the in situ $CO_2$ observations by the PICARRO instrument are well reproduced by WRF-GHG, resulting in a correlation
coefficient of 0.75 and a very small mean bias error of $-0.15$ ppm (see Table 3). As at STD, the diurnal $CO_2$ cycle at MA shows
a daytime minimum and a nighttime maximum (see Fig. 14a), however the amplitude is much smaller. This pattern is caught
by WRF-GHG, although the amplitude is slightly underestimated. During the day, WRF-GHG shows a small overestimation of
the $CO_2$ measurements of about 0.9 ppm, while at night a slight underestimation (about 0.4 ppm) is found. As seen on Fig. 14b,
the modeled diurnal variation is almost entirely produced by the biogenic tracer, indicating that the biogenic flux calculated by
VPRM might be the reason for the model-observations discrepancies. The VPRM parameters used in the model are based on
model tests in the Amazon region.





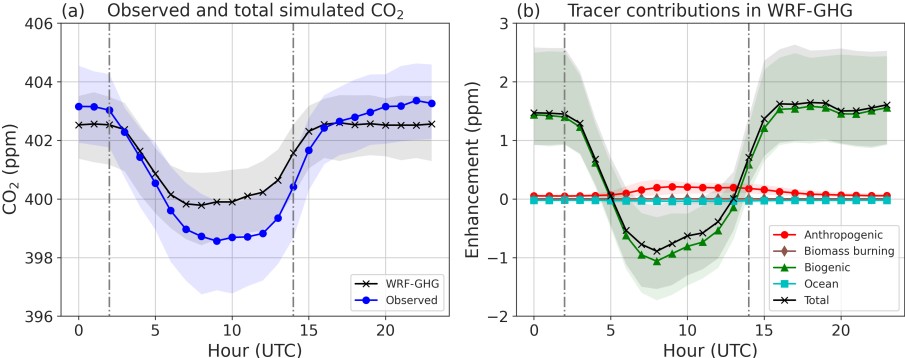

**Figure 14.** Same as Figure 7 but for Maïdo.

Foucart et al. (2018) showed that due to surface radiative cooling, the observatory is primarily situated in the free troposphere at night. The air at MA is then disconnected from local pollution sources and air from remote regions can be sampled. Indeed, no anthropogenic contribution is detected during the nighttime. However, the observed and simulated diurnal cycle of $CO_2$ (Fig. 14a and b), shows that nighttime measurements at MA are still influenced by respiration of the local vegetation.

The anthropogenic contribution at MA is very minor in WRF-GHG, which is expected because of the remote location of the

Observatory. A very small enhancement is identified during the day. Since the local grid cell used for the model comparison does not include any anthropogenic flux, this enhancement is advected from elsewhere. It has been shown that orographic lifting can bring polluted air from coastal areas in the West towards MA during the day (Foucart et al., 2018; Duflot et al., 2019). Despite that these westerly winds during the day were not reproduced by WRF-GHG (see Sect. 4.1.1), a daytime anthropogenic enhancement is found in the simulations. The model components representing biomass burning and ocean fluxes at MA are

negligible.

So, according to WRF-GHG the main contribution (above the background) to the $CO_2$ signal at MA is coming from the local vegetation and its photosynthesis and respiration, leading to a distinct diurnal cycle. The importance of the surrounding biosphere for the surface observations at MA was also found by Verreyken et al. (2021) for volatile organic compounds. Even though the diurnal cycles of $CO_2$ at STD and MA display similar patterns of minima during the day and maxima at night, they

are caused by entirely different mechanisms.

### 4.3.2  Surface $CH_4$

Because of the importance of accurate background concentrations, the model performance at simulating in situ $CH_4$ concentrations at MA is very similar compared to STD: the correlation is low (0.30) and the model overestimates the observations by circa 19 ppb. The modeled signal consists almost entirely out of the anthropogenic tracer (additional to the background

signal), see Fig. 15a. Since the errors at MA follow the same pattern (Fig. 12) as at STD, and because inaccurate background information affects all $CH_4$ simulations, the seasonal bias in the CAMS reanalysis is also the cause for the weak model per-





formance at MA. The errors at MA are larger than those at STD, likely due to the relatively low resolution of the EDGAR inventory ($0.1°$) used for anthropogenic $CH_4$ emissions, leading to horizontal dilution: the concentration difference between high emission areas and their surroundings becomes smaller leading to an overestimation of the emissions in the low emission areas. In contrast with the model results, no diurnal cycle could be detected in the observations (not shown).

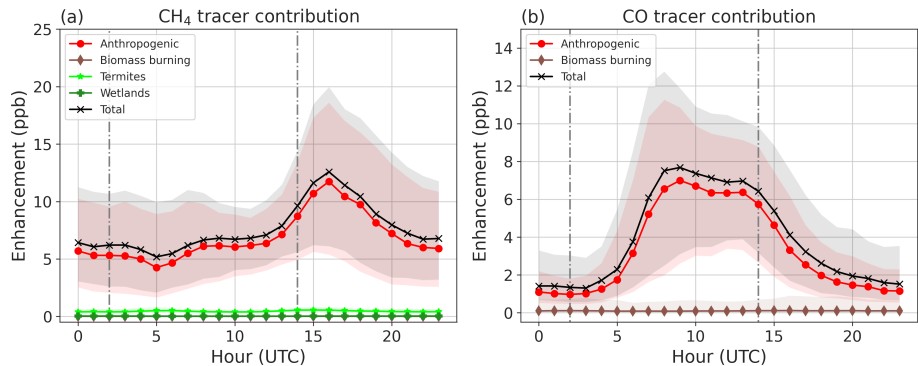

**Figure 15.** Diurnal cycle of tracer contributions for (a) $CH_4$ and (b) CO in situ surface concentration at MA.


### 4.3.3 $XCH_4$

The model-data comparison for the NDACC data shows a very weak correlation ($0.37$) and a model underestimation of NDACC $XCH_4$ ($-5.65$ ppb). This mean bias error has an opposite sign compared to TCCON $CH_4$. The difference between NDACC and TCCON $XCH_4$ is mainly due to their difference in vertical sensitivity (Zhou et al., 2018). This pattern is the same as the

one found by Ramonet et al. (2020) in comparisons of the CAMS reanalysis with NDACC and TCCON $CH_4$. Again, a seasonal pattern is found in the relative differences (Fig. 10b), caused by the reported bias for $CH_4$ in the CAMS reanalysis data.

The tracer contributions to the $XCH_4$ signal at MA in WRF-GHG are very similar to those at STD, with seasonal enhancements from biomass burning and wetlands from Africa, alongside a more constant anthropogenic part (Fig. 11b). Remark that the contributions at MA seem to be slightly larger than those at STD. This is because the atmospheric column above the high-

altitude station of MA is smaller than the one above STD and because the enhancements are transported from Africa and Madagascar by the westerlies higher up in the troposphere. The relative contributions averaged over the column are then higher at MA than at STD.

### 4.3.4 Surface CO

WRF-GHG captures the in situ surface CO time series at MA quite well: there is a high correlation of $0.83$ (Table 3, Fig.

6e). The RMSE is about $11$ ppb and there is a small model overestimation of $5.51$ ppb. During the day, a small anthropogenic enhancement is found (see Fig. 15b), as for $CO_2$. As already noticed, thermal contrasts make air masses from the coastal areas arise during the day along the mountain slope before reaching the Maïdo Observatory. This air contains anthropogenic





pollution. At the observatory itself no anthropogenic CO fluxes are implemented, so the model is representing this daytime advection to some extent.

Another contributor is the BB signal from August to December. The contribution is not very visible in the diurnal cycle due to its seasonal nature, but daily enhancements of up to 40 ppb are simulated by WRF-GHG. BB contributions from the African continent and Madagascar are highest during the night, when MA is generally located in the free troposphere and transport from distant regions is detected (Baray et al., 2013).

### 4.3.5 XCO

A very high correlation (0.90) is found for the hourly averaged paired column data of NDACC and WRF-GHG (Table 3). In general WRF-GHG slightly overestimates the observed XCO (MBE of 1.81 ppb). Remark that the errors are larger (the overestimation is larger) for the TCCON data compared to the NDACC data (also Fig. 10b). This is probably linked to the fact that TCCON XCO is generally a few ppb lower than NDACC XCO, due to differences in the retrieval algorithms and data corrections within the two networks (Zhou et al., 2019).

As for $XCH_4$, the average monthly tracer contributions of XCO at MA are very similar as those at STD (Fig. 11b), with large contributions of BB emissions in the months August to December. Because of the unique location of MA, both the in situ observations and the column-averaged XCO are sensitive to these large seasonal events.

## 5 Model resolution

The above analysis was done using the WRF-GHG simulations from the innermost domain d03 (Fig. 2), which has a horizontal
resolution of 2 by 2 km. As surface in situ observations are heavily influenced by local fluxes and dynamics, this high resolution is necessary to represent these measurements accurately, especially in regions with complex topography. As the ground-based remote-sensing FTIR observation sample a much larger volume of air, a lower model resolution is likely sufficient to catch the fluxes and processes that influence them. Therefore, the model-data comparison for domains d01 and d02, with a horizontal resolution of 50 km and 10 km, respectively, is given in this section. Table 4 gives the statistical metrics for the FTIR observa-
tions at both sites, for all model domains.

The results are very similar among the different model resolutions, indicating that even a horizontal resolution of 50 km (as in d01) could be sufficient to simulate the FTIR observations at Reunion Island. In addition to the larger sampling volume, this can be explained by the fact that the most important contributions to the column are coming from remote areas such as Africa and Madagascar, situated in d01 and d02. The added value of high-resolution transport in d03 is negligible for the FTIR
observations.





| | STD | | | | | | | | | MA | | | | | |
|---|---|---|---|---|---|---|---|---|---|---|---|---|---|---|---|
| | XCO$_2$ (ppm) | | | XCH$_4$ (ppb) | | | XCO (ppb) | | | XCH$_4$ (ppb) | | | XCO (ppb) | | |
| | d01 | d02 | d03 | d01 | d02 | d03 | d01 | d02 | d03 | d01 | d02 | d03 | d01 | d02 | d03 |
| RMSE | 0.66 | 1.26 | 0.75 | 12.45 | 11.89 | 10.26 | 8.05 | 8.01 | 8.08 | 10.55 | 11.03 | 10.80 | 7.64 | 7.24 | 7.37 |
| MBE | 0.12 | -0.24 | -0.37 | 8.69 | 6.89 | 5.69 | 5.06 | 5.02 | 5.07 | -5.22 | -6.00 | -5.65 | 1.92 | 1.46 | 1.81 |
| CORR | 0.90 | 0.75 | 0.90 | 0.27 | 0.34 | 0.31 | 0.88 | 0.89 | 0.89 | 0.36 | 0.37 | 0.37 | 0.89 | 0.89 | 0.89 |

**Table 4.** Overview of WRF-GHG performance of simulating hourly FTIR observations of GHG at Reunion Island, for all model domains. Comparison with the columns observations is done using the smoothed model profiles.

## 6    Conclusions

We studied the variability of $CO_2$, $CH_4$ and CO surface and column observations at Reunion Island and evaluated possible factors influencing their observed mole fractions. This was achieved by comparing the available data sets with simulations of the WRF-GHG model over two periods between 2015 and 2017, totalling 20 months. The model performance was first evalu-
ated for basic meteorological fields both near the surface as well as along atmospheric profiles. WRF-GHG shows good skill in reproducing these measurements, especially temperature. However, the local wind speed in Saint-Denis is overestimated by almost $4\ \mathrm{ms^{-1}}$ and also at Maïdo, there are some discrepancies in the wind speed and direction, which are likely linked to the complex topography and the model resolution of 2 km not being sufficient to represent very local dynamical processes.

Nevertheless, several valuable insights were identified by analyzing the results. The surface $CO_2$ mole fractions in Saint-Denis
follow a distinct diurnal cycle with values up to 450 ppm at night, driven by local anthropogenic emissions, planetary boundary layer dynamics and accumulation due to low wind speeds. Additionally, the signal includes respiration from vegetation that is carried by eastern winds from more rural regions. Due to the overestimation of local wind speeds in the capital, WRF-GHG underestimates the nocturnal $CO_2$ buildup, leading to only a correlation coefficient of 0.62.

At the Maïdo Observatory on the other hand, a similar diurnal cycle of $CO_2$ is found but with much smaller amplitude. There,
the surface $CO_2$ mole fractions are essentially driven by the surrounding vegetation that take up $CO_2$ during the day and release $CO_2$ during the night through respiration. A small underestimation of the diurnal cycle amplitude in WRF-GHG might indicate that the VPRM parameters could be improved for this region.

A high correlation of 0.9 was found between the hourly column $XCO_2$ values of WRF-GHG and the corresponding TCCON observations. These column-averaged mole fractions describe different air masses than those near the surface. As shown by
previous studies, these measurements are influenced by processes from distant areas such as Africa and Madagascar. The different model tracers show contributions from fire emissions during the biomass burning season, but also positive biogenic enhancements associated with the dry season.

WRF-GHG fails to reproduce accurately the different $CH_4$ observations at Reunion Island due to a seasonal bias in the background arising from the CAMS reanalysis. Furthermore, tracer contributions reveal that the emission sources within the model
domain have only a minimal effect on the overall signal. Besides the background, local anthropogenic fluxes are the major





| | Evergreen forest | Deciduous forest | Mixed forest | Shrubs | Savanna | Crops | Grasses |
|---|---|---|---|---|---|---|---|
| $PAR_0$ | 993.9 | 324.0 | 206.0 | 303.0 | 6860.7 | 2329.0 | 15475.5 |
| $\lambda$ | 0.1096 | 0.1729 | 0.2555 | 0.0874 | 0.0277 | 0.0417 | 0.0568 |
| $\alpha$ | 0.2114 | 0.3258 | 0.3422 | 0.0239 | -0.2535 | -0.0814 | -0.3122 |
| $\beta$ | 1.8187 | 0. | 0. | 0. | 7.1125 | 3.6716 | 7.3377 |

**Table A1.** VPRM parameters used within WRF-GHG

source influencing the in situ observations at Reunion Island. However in Saint-Denis, and even more so at Maïdo, the anthro-pogenic $CH_4$ emissions from EDGAR are likely largely overestimated.

Again, impacts from Africa and Madagascar can be seen in the FTIR observations: fire plumes during the biomass burning season and wetland emissions during the rainy season.

The WRF-GHG model is able to simulate the CO levels at Reunion Island with a relative high degree of accuracy. For XCO, the importance of biomass burning plumes from Africa and elsewhere for the observed variability is confirmed. The in situ observations at Maïdo can detect anthropogenic signals from the coastal regions during the day and biomass burning enhance-ments from afar at night, when the Observatory is located in the boundary layer and the free troposphere, respectively. Despite the differences in modeled and observed wind directions, WRF-GHG is able to mimic to some extent the anabatic winds that

are typical for the northwest part of the island.

The high model resolution of 2 km is needed to accurately represent local fluxes and small-scale processes that affect the in situ observations. Because of the complex topography and the unique local wind patterns, an even higher resolution might be needed to simulate more precisely the observations at Maïdo. On the contrary, to simulate the column FTIR observations, a model resolution of 50 km appears to be sufficient. Note that this may not be the case in areas where the most significant factors

influencing the measured signal are nearer.

This study showed an application of the WRF-GHG model in a region of the globe where it had not yet been run before. It demonstrated that WRF-GHG had great skill in simulating the meteorological fields and different in situ surface and column observations of GHG. However results are highly dependent on accurate boundary conditions and the availability of high resolution emission inventories.

**Appendix A: VPRM parameters**

As mentioned in Sec. 3.1, this study uses the VPRM parameter set that was optimized by Botía et al. (2021) for the Amazon region in Brazil. Table A1 gives the exact values for every vegetation class.





## Appendix B: Smoothing model data

A smoothing correction is applied when comparing the model data with the TCCON and NDACC data. Retrieved column-

averaged mole fractions are affected by the observing system characteristics and therefore Rodgers and Connor (2003) suggest to take into account the a priori information and averaging kernels of the retrieval when calculating the Xgas of the model. The different steps undertaken to calculate this are explained hereafter for TCCON and NDACC separately.

Generally, the smoothed Xgas from WRF-GHG is calculated as:

$$X_{\text{gas},s} = \frac{TC_s^{\text{gas}}}{TC_{\text{air}}} = \frac{\sum_i PC_s^{\text{gas},i}}{TC_{\text{air}}} = \frac{\sum_i x_s^i PC_{\text{air}}^i}{TC_{\text{air}}}, \tag{B1}$$

where $PC_{\text{air}}^i$ is the partial column number density of dry air in layer $i$ and $x_s^i$ is the volume mixing ratio with respect to dry air in layer $i$ of the smoothed model profile. In the following, all parameters indicating a volume mixing ratio or column number density are also with respect to dry air, however for brevity it will not be specified any more.

### B1   TCCON

Equation (B1) requires a smoothed vertical profile ($x_s^i$). Since TCCON does not provide profile retrievals, we can not calculate

this. Instead, we use the following smoothing equation for TCCON:

$$X_{\text{gas},s} = \frac{TC_{\text{apriori}}^{\text{gas}}}{TC_{\text{TCCON}}} + \boldsymbol{a} \cdot \left( \frac{\boldsymbol{PC}_{\text{WRF,regrid}}^{\text{gas}}}{TC_{\text{WRF,regrid}}} - \frac{\boldsymbol{PC}_{\text{apriori}}^{\text{gas}}}{TC_{\text{TCCON}}} \right), \tag{B2}$$

where $TC_{\text{TCCON}}$ is the total column (number density) of dry air from TCCON (for an atmospheric column up to 50 hPa). Similarly, $TC_{\text{apriori}}^{\text{gas}}$ is the total column of the a priori mole fraction from TCCON calculated as the sum of the partial columns $PC_{\text{apriori}}^{\text{gas},i}$ over those layers $i$ that are below 50 hPa. Further, $\boldsymbol{a}$ is the vector with the column averaging kernels of TCCON.

The regridded partial column profile of WRF-GHG ($\boldsymbol{PC}_{\text{WRF,regrid}}^{\text{gas}}$) and the total column of dry air from WRF ($TC_{\text{WRF,regrid}}$) are calculated in a few steps which are explained below. By including the total column of dry air from WRF in eq. (B2), we want to eliminate potential differences in air between TCCON and the model. As such, the priority is given to the volume mixing ratio profiles (instead of the calculation of dry air).

1. Extend the WRF-GHG atmospheric profiles (gas mole fraction, pressure, temperature and water vapour) above the model

limit (50 hPa) using information of the TCCON a priori profiles.

2. Calculate the dry-air partial column in layer $i$ using the ideal gas law:

$$PC_{\text{air}}^i = \frac{P^i}{RT^i} \frac{\tau^i}{1 + 1.6075 q^i},$$

with $P$ atmospheric pressure, $T$ air temperature, $R$ ideal gas constant, $q$ mass mixing ratio of water vapour and $\tau$ layer thickness.

3. Calculate the gas number density partial columns as $PC_{\text{gas}}^i = x_{\text{gas}}^i PC_{\text{air}}^i$, with $x_{\text{gas}}^i$ the gas mole fraction in layer $i$.





4. Regrid these partial column profiles to the full TCCON grid using a transformation matrix $\mathbf{D}$ as in Langerock et al. (2015):

$$PC_{\text{WRF,regrid}}^{\text{gas}} = \mathbf{D} \cdot PC_{\text{gas}} \text{ and } PC_{\text{WRF,regrid}} = \mathbf{D} \cdot PC_{\text{air}}$$

5. Finally, $TC_{\text{WRF,regrid}} = \sum_i PC_{\text{WRF,regrid}}^i$ where the sum is taken over all layers $i$ below 50 hPa.

**B2 NDACC**

The smoothed Xgas from WRF-GHG at MA is calculated slightly different than at STD, as for NDACC volume mixing ratio profiles are provided. The smoothing equation can be written as:

$$x_{\text{gas},s} = x_{\text{apriori}}^{\text{gas}} + \mathbf{A} \cdot \left( x_{\text{WRF,regrid}}^{\text{gas}} - x_{\text{apriori}}^{\text{gas}} \right), \tag{B3}$$

where $x_{\text{apriori}}^{\text{gas}}$ is the volume mixing ratio (vmr) a priori profile from NDACC and $\mathbf{A}$ is the NDACC vmr averaging kernel matrix.
Similar as for TCCON, a few steps need to be made to make the WRF-GHG data fit in eq. (B3). Steps 1-4 as described above should be followed, but using NDACC information instead of TCCON (a priori vmr profile, temperature, and water vapour profiles, vertical grid). Then the regridded vmr profile from WRF-GHG is calculated as $x_{\text{WRF,regrid}}^{\text{gas}} = \dfrac{PC_{\text{WRF,regrid}}^{\text{gas}}}{PC_{\text{WRF,regrid}}}$. Finally, the smoothed dry-air mole fraction at MA is given by:

$$X_{\text{gas},s} = \frac{\sum_i x_{\text{gas},s}^i PC_{\text{WRF,regrid}}^i}{\sum_i PC_{\text{WRF,regrid}}^i},$$

where the sum is taken over all layers $i$ below 50 hPa.

**Appendix C: Time series**

The full time series of both the observed and modeled concentrations at STD and MA are given in the figures hereafter. Figure C1 and C2 show the time series of the in situ data at STD and MA, respectively. Similarly, Fig. C3 and C4 show the comparison of the FTIR data at STD (TCCON) and MA (NDACC).

*Author contributions.* SC set up the model simulations, made the analysis and wrote the manuscript. JB provided the anthropogenic CO fluxes at Reunion Island and its description. VD provided the meteorological radiosonde profiles. BL provided expertise on the FTIR data and model comparison techniques. DF assisted in setting up the model and together with JFM helped with correctly interpreting the results. JMM, CH and NK provided the in situ measurements on Reunion Island. EM and MDM provided general guidance and support during the analysis and revised and edited the manuscript. All authors reviewed and commented on the manuscript.

*Competing interests.* The authors declare that they have no conflict of interest.





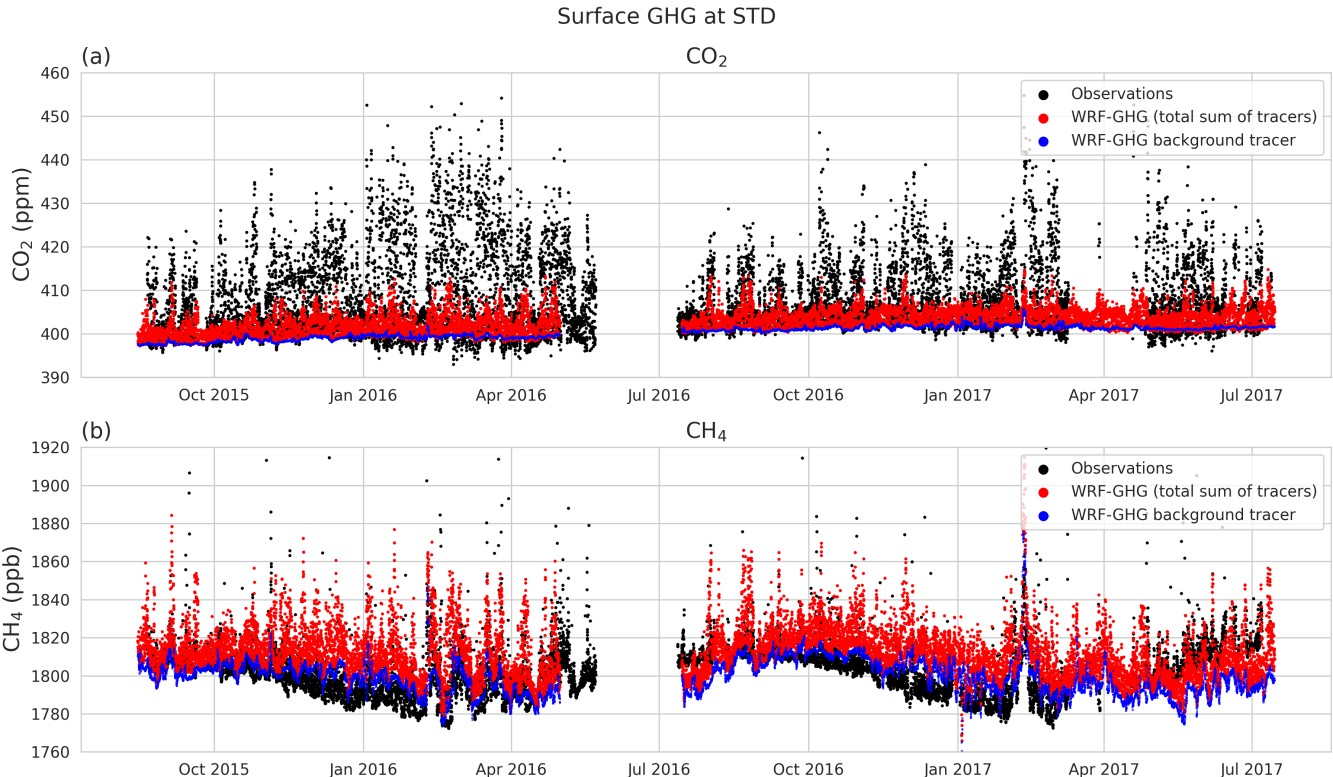

**Figure C1.** Time series of all observed (black) and modeled (red) in situ concentrations at STD of (a) $CO_2$ and (b) $CH_4$. The blue dots represent the modeled background tracer.

*Acknowledgements.* We acknowledge the providers of the observational data sets. The data used in this publication were obtained from Martine De Mazière as part of the Network for the Detection of Atmospheric Composition Change (NDACC) and are available through the NDACC website www.ndacc.org. The TCCON data were obtained from the TCCON Data Archive hosted by CaltechDATA at https://tccondata.org (https://doi.org/10.14291/TCCON.GGG2014.REUNION01.R1). The ERA5 and CAMS reanalysis dataset (Hersbach et al. (2018a, b) used as

input for the WRF-GHG simulations, were downloaded from the Copernicus Climate Change Service (C3S) Climate Data Store. The results contain modified Copernicus Atmosphere Monitoring Service information 2020-2021. Neither the European Commission nor ECMWF is responsible for any use that may be made of the Copernicus information or data it contains.

The authors also acknowledge the European Communities, the Région Reunion Island, CNRS, and Université de la Reunion Island for their support and contribution in the construction phase of the research infrastructure OPAR (Observatoire de Physique de l'Atmosphère à La

Réunion). OPAR is presently funded by CNRS (INSU), Météo France, and Université de La Réunion, and managed by OSU-R (Observatoire des Sciences de l'Univers à La Réunion, UAR 3365). OPAR is supported by the french research infrastructure ACTRIS-FR (Aerosols, Clouds, and Trace gases Research InfraStructure - France).

We thank Christophe Gerbig, Roberto Kretschmer and Thomas Koch (MPI BGC) for their work on the VPRM preprocessor and support on installing the software at the BIRA-IASB servers. The authors also wish to thank Julia Marshall (DLR) and Michael Gałkowski (MPI BGC)

for their guidance in running the WRF-GHG model and interpreting its results. Additionally, we thank the broader WRF-GHG community for



**Figure C2.** Time series of all observed (black) and modeled (red) in situ concentrations at MA of (a) $CO_2$, (b) $CH_4$ and (c) CO. The blue dots represent the modeled background tracer.



**Figure C3.** Time series of all observed (black) and modeled (red) column concentrations at STD of (a) $CO_2$, (b) $CH_4$ and (c) CO. The blue dots represent the modeled background tracer. The modeled data is hourly and smoothed. The observed data is scaled to the atmospheric column until 50 hPa and all available measurements are shown.

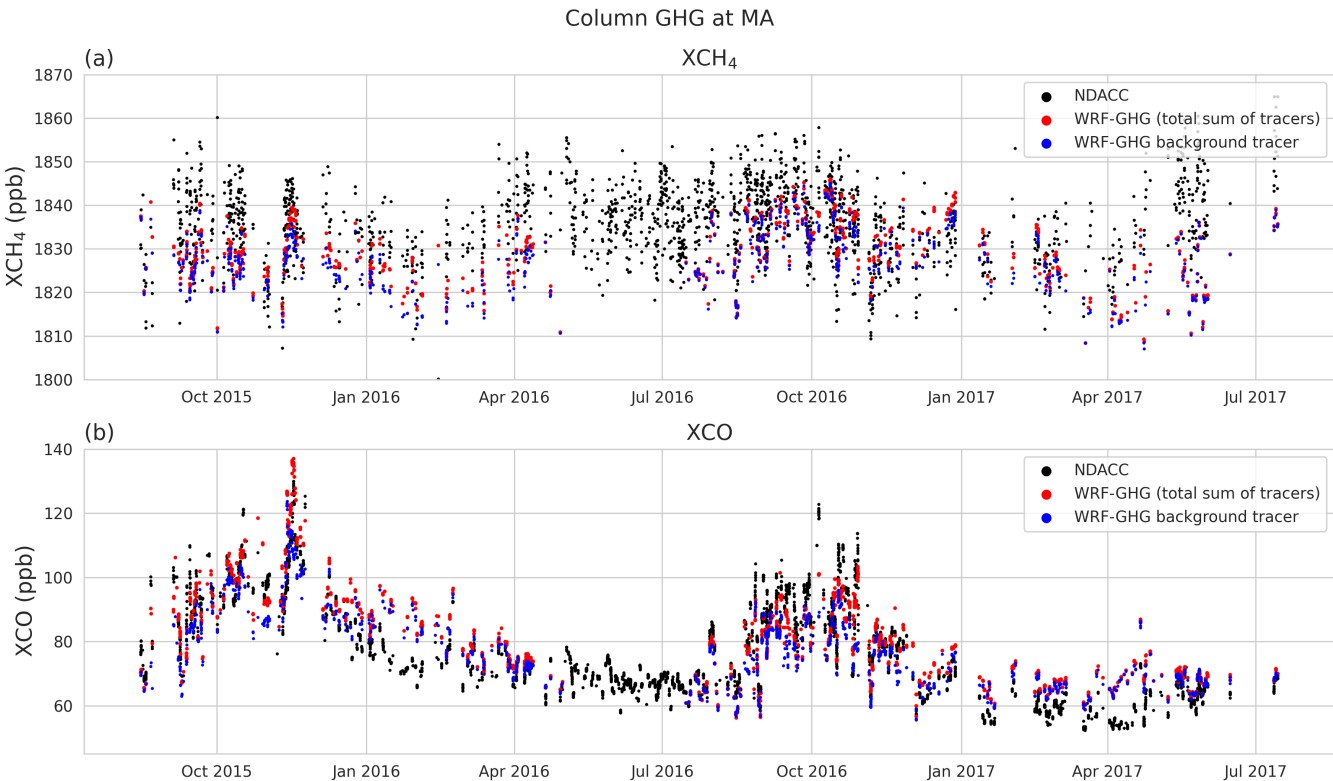

**Figure C4.** Time series of all observed (black) and modeled (red) column concentrations at MA of (a) CH$_4$ and (b) CO. The blue dots represent the modeled background tracer. The modeled data is hourly and smoothed. The observed data is scaled to atmospheric column until 50 hPa and all available measurements are shown.

regular exchange of expertise. Finally, we thank Mahesh Kumar Sha (BIRA-IASB) and Minqiang Zhou (IAP, CAS) for helpful discussions on the interpretation of the data.



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
