# Peer review of "Analysis of CO2, CH4 and CO surface and column concentrations observed at Reunion Island by assessing WRF-Chem simulations"

_Atmospheric Chemistry and Physics, 2022_

## Referee Comment (RC1)

The paper under consideration entitled "Analysis of $CO_2$, $CH_4$ and CO surface and column concentrations observed at Reunion Island by assessing WRF-Chem simulations" by S. Callewaert et al. presents in situ and remote sensing data collected at two sites on Reunion Island during two periods in the time frame 2015 to 2017. The authors perform an insightful comparison of the measurements with WRF-GHG simulations.

I have mainly minor comments (listed below). Overall, the paper is very readable: the applied methods are appropriate and the treatment of topics is well structured. Nevertheless, I would recommend that the discussion of those findings the authors would assign the highest importance should be enhanced. Currently, the conclusions read like a long listing of various minor items – all correct, most of them interesting and worth to mention – but altogether leaving the reader somewhat with the impression that the study lacks a scientific focus.

Comments in detail:

Introduction:

Paragraph starting line 33 – this paragraph details a bit on strengths and weaknesses of remote sensing (NDACC and TCCON) and in-situ measurements. The more recent COCCON network should be mentioned here, as it aims to fill in the gap between global and small-scale measurements (allowing arrangements of portable spectrometers for observing dedicated areas of interest as, e.g., metropolitan areas). See

https://amt.copernicus.org/articles/8/3059/2015/

https://acp.copernicus.org/articles/19/3271/2019/

https://amt.copernicus.org/articles/14/1047/2021/

line 54: "near the surface, winds … originate in" -> "near the surface, air masses … originate in"

Line 59: "However …" -> "However, …"

Line 69ff: Here, WRF studies in the context of city emissions conducted by other groups should be mentioned here as well, e.g.,

https://acp.copernicus.org/articles/19/11279/2019/

https://acp.copernicus.org/articles/21/13131/2021/acp-21-13131-2021.pdf

Paragraph starting line 128: this explanation has some redundancy (FTIR observations … providing mole fractions in an atmospheric column … . The spectra are used … to retrieve the total column-averaged dry-air …)

Paragraph starting at line 147: The authors correctly stressed earlier the tying of TCCON results to WMO standards. The systematic error budget of NDACC vs TCCON should be mentioned here explicitely (as both data sets are mixed together later in the discussion) – which amount of bias between the two remote sensing data sets might be expected?

Line 309: "agree less" -> "agree less well"

Figure 6 and associated discussion: Given the elevated altitude of the MA station, it might be worth to appropriately filter the in-situ data (selecting those data which are sampling free tropospheric air) and specifically discuss the resulting subset, see a study following this approach for the Izana station on Tenerife here: https://amt.copernicus.org/articles/7/2337/2014/

Para starting line 373 discussing long-range transport. The findings reported by Frey et al. for the Gobabeb station (see section "influence of African biosphere") might be of special relevance here: https://amt.copernicus.org/articles/14/5887/2021/.

Statement line 380: Does this imply that remote sensing measurements are more useful than in situ measurements for studying long-range transport (possibly also reflected by the fact that lower model resolution is sufficient for achieving good correlation between model and measurements)?

Line 391: "in more details" -> "in more detail"

---

## Referee Comment (RC2)

Review of Callewaert et al. 2022: "Analysis of CO2, CH4 and CO surface and column concentrations observed at Reunion Island by assessing WRF-Chem simulations" in ACPD.

Callewaert et al. present a comparison of modelled and observed in-situ dry air mole fraction and the total column-averaged dry-air mole fraction of CO2, CO and CH4 at two sites on Reunion Island (St-Denis and Maido observatory). The atmospheric composition is modelled using a nested version of WRF-GHG with different emission priors, while observations are performed using cavity ring-down spectrometers and solar tracking FTIR systems.

The study demonstrates a good ability for WRF-GHG to reproduce atmospheric temperature and a limited ability to reconstruct atmospheric wind patterns (with a significant high bias in wind speed). Total column and in-situ observations correlate well for atmospheric CO and CO2, while CH4 data shows a surprisingly low correlation coefficient.

Overall, the paper is well written and nicely structured, so readers can follow the logic of the comparison. It presents and discusses atmospheric data from a chronically under-sampled region (Indian Ocean) with the aid of a state of the art atmospheric transport model. The analysis is sound and the scope of the paper well suited for ACP and its readers. After addressing the suggestions and technical correction below, I can fully recommend the paper for publication.

General comments:

Some results warrant a more detailed discussion, especially the issue of the low correlation of $CH_4$. Looking at Figure 6 (b) and (d) seems to suggest that there could be two apparent distributions for $CH_4$ that, if fitted separately, could produce much more reasonable slopes and improved coefficients. Have you attempted to separate the data based on external drivers that could explain the two distributions?

[Figure]

Minor and technical comments:

L1: The authors should consider changing the title as they report modelling result using WRF-GHG (passive tracer) rather than the version of WRF with active chemistry (WRF-CHEM).

L14: please add that the Pearson's correlation coefficient was used here.

L67: please correct "etc…"

L83: please elaborate on what "…" refers to or just give the elements in the brackets as an example.

L225, Figure 2: please consider adding the information on the vertical resolution and top of the domain in the caption of Figure 2

L309: please change to "… agree less well …"

L337: Please elaborate on the assumption that there is "no vegetation within the city".

A simple search of aerial photos of St. Denis reveals multiple parks and vegetation along the shoreline. Maybe the assumption is rather that the impact of local vegetation is negligible compared to fossil fuel combustion?

L368: Is the statement related to the nighttime respiration true in general or here specifically for a local imbalance in the boundary layer.

---

## Author Comment (AC1)

**Final response to comments on "Analysis of CO2, CH4 and CO surface and column concentrations observed at Reunion Island by assessing WRF-Chem simulations"**

First and foremost, we want to express our gratitude to the referees for taking the time to read our article and provide constructive feedback. We will reply to their comments below. The comments given by the referees are written in black, while author comments are in blue. Modifications to the manuscript are written in *italic*.

**Referee 1**

**Main comment:**

I would recommend that the discussion of those findings the authors would assign the highest importance should be enhanced. Currently, the conclusions read like a long listing of various minor items – all correct, most of them interesting and worth to mention – but altogether leaving the reader somewhat with the impression that the study lacks a scientific focus.

We propose to rewrite the complete conclusion section such that the questions from the introduction are explicitly addressed. Below is our alternative conclusion (to replace lines 529 - 560).

[revised manuscript text omitted]

**Minor comments:**

Paragraph starting line 33 – this paragraph details a bit on strengths and weaknesses of remote sensing (NDACC and TCCON) and in-situ measurements. The more recent COCCON network should be mentioned here, as it aims to fill in the gap between global and small-scale measurements (allowing arrangements of portable spectrometers for observing dedicated areas of interest as, e.g., metropolitan areas). See

https://amt.copernicus.org/articles/8/3059/2015/

https://acp.copernicus.org/articles/19/3271/2019/

https://amt.copernicus.org/articles/14/1047/2021/

The following sentence was added into this paragraph at line 41: "Recently, this kind of observations from mobile low-cost FTIR spectrometers within the Collaborative Carbon Column Observing Network (COC-CON) has been used to constrain fluxes in urban regions (Hase et al. (2015), Makarova et al. (2021), and Vogel et al. (2019))."

line 54: "near the surface, winds . . . originate in"  $\rightarrow$  "near the surface, air masses . . . originate in" Done

Line 59: "However ...."  $\rightarrow$  "However, ...." Done

Line 69ff: Here, WRF studies in the context of city emissions conducted by other groups should be mentioned here as well, e.g.,

https://acp.copernicus.org/articles/19/11279/2019/

https://acp.copernicus.org/articles/21/13131/2021/acp-21-13131-2021.pdf

The first article by Zhao et al. (2019) suggested by the referee is already mentioned in the text (line 71). The second article by Jones et al. (2021) which assesses urban methane emissions using FTIR observations is indeed very interesting. However, the study uses the Stochastic Time-Inverted Lagrangian Transport (STILT) model and not WRF. Therefore, it won't be added to the references in the introduction.

Paragraph starting line 128: this explanation has some redundancy (FTIR observations ... providing mole fractions in an atmospheric column ... . The spectra are used ... to retrieve the total column-averaged dry-air ...)

It is correct that both sentences are very similar. However, they were written for a different purpose.

The first sentence aims to indicate in general the difference with respect to the in situ measurements: the FTIR observations measure a signal along the solar path of an atmospheric column. The second sentence aims to be more specific about the product that is retrieved from these FTIR observations and further used in this study: the total column-averaged dry-air mole fraction (the so-called Xgas).

We deleted the first sentence in the manuscript and additionally mentioned the location of the FTIR instrument next to the PICARRO analyzer. The paragraph starting at line 129 now becomes:

"In September 2011, BIRA-IASB installed a high-resolution Bruker IFS 125HR FTIR at STD, next to the PICARRO analyzer. This instrument is primarily dedicated to measuring the near-infrared (NIR: 4000 - 16000 cm-1) spectra and contributes to TCCON (Wunch et al. (2011)). The solar spectra are used to retrieve the total column-averaged dry-air mole fraction of  $CO_2$ ,  $CH_4$  and CO (De Maziere et al. (2017))".

Paragraph starting at line 147: The authors correctly stressed earlier the tying of TCCON results to WMO standards. The systematic error budget of NDACC vs TCCON should be mentioned here explicitly (as both data sets are mixed together later in the discussion) – which amount of bias between the two remote sensing data sets might be expected?

The differences between NDACC and TCCON data are mentioned later in the article, in the results section. First the TCCON results are discussed, and afterwards the NDACC results are examined. At the latter, we tried to refer back to the TCCON results and justify the observed differences. For XCH4, this is done at lines 478-479 by referring to Zhou et al. (2018). They showed that XCH4 at Reunion Island from TCCON is about 10 ppb lower than NDACC XCH4 due to differences in vertical sensitivity. A more comprehensive study about the differences between TCCON and NDACC including 6 sites was made for XCO by Zhou et al. (2019). This is referred to on lines 502-504. They found that for XCO the bias between the two networks was below 2 % in the Southern Hemisphere, and that taking into account the smoothing error is important. More specifically, a systematic bias of 2.5 % was found between NDACC XCO and TCCON XCO at Reunion Island. We will adapt these lines in the article to be more explicit: Lines 478-479: "Zhou et al. (2018) showed that NDACC XCH4 is generally about 10 ppb lower than TCCON XCH4 at Reunion Island due to their difference in vertical sensitivity."

Lines 502-504: "This is probably linked to biases between the TCCON and NDACC data sets. Zhou et al. (2019) showed that there is a bias of 2.5% between TCCON XCO and NDACC XCO at Reunion Island, due to differences in the retrieval algorithm and data corrections."

Line 309: "agree less"  $\rightarrow$  "agree less well" Done

Figure 6 and associated discussion: Given the elevated altitude of the MA station, it might be worth to appropriately filter the in-situ data (selecting those data which are sampling free tropospheric air) and specifically discuss the resulting subset, see a study following this approach for the Izana station on Tenerife here: https://amt.copernicus.org/articles/7/2337/2014/

The study of Sepúlveda et al. (2014) mentioned by the referee compares the  $CH_4$  mole fractions of several NDACC sites with co-located in situ observations, which are first filtered to represent regional-scale signals (of the free troposphere). The proposed filter for Izaña (which is a subtropical high mountain observatory, like Maïdo) is averaging the nighttime in situ observations to obtain a nighttime mean. The solar FTIR measurements, which can only be performed during the day, are averaged to obtain a daily mean mole fraction. This daily mean is then compared with the mean of the two in situ nighttime averages around that day.

We can recalculate the statistical metrics and remake the correlation plots of Figure 6 in the manuscript for the nighttime data at Maïdo, as suggested by the referee. Like this, the model performance of those observations that should be representative of the free troposphere can be evaluated. Figures A1, A2 and A3 here below show these results. For CO, the model performs equally well for all observations (day or night): both the root mean square error (RMSE) and mean bias error (MBE) show similar values when taking only the nighttime pairs (compared to all or only daytime). The correlation coefficients are all equal to 0.83.

As mentioned in the manuscript, a low correlation coefficient and larger errors are found for the model performance of  $CH_4$  because of a seasonal bias in the background information. Here again, similar statistics are found when comparing daytime observations, nighttime observations or both together. However, the metrics seem to be slightly better for the daytime points compared to the nighttime points. This might be related to the incorrect spike at night in the anthropogenic tracer of WRF-GHG as shown in Figure 15(a) in the manuscript, linked to EDGAR emissions. It is not clear why WRF-GHG simulates higher anthropogenic values for nighttime points compared to daytime points, leading to a larger overestimation.

The plots for  $CO_2$  in Fig. A3 also show similar statistics for all subsets. There is a small overestimation of the daytime points and a small underestimation of the nighttime points. In the manuscript we linked this to the biogenic tracer and suggested this might be caused by inaccurate VPRM parameters. The higher correlation during the day might suggest that the photosynthesis parameters are more representative than the respiration parameters for the biosphere at Reunion Island. Because of the large impact of local surface fluxes on  $CO_2$ , the nighttime in situ observations at Maïdo are less representative of the free troposphere than the other two gases.

We added a few sentences in the manuscript (around line 440) clarifying that the analysis was made on the full dataset because no significant differences were found on any subset:

"Remark that all statistical analyses of in situ observations at MA are performed on the complete dataset. Studies at other high-altitude stations often filter only those measurements which are representative for the free troposphere (Sepúlveda et al. (2014)). However, analyses comparing only day- or nighttime data at MA showed no significant differences in the results."

Figure A1: Scatter plot of observed and simulated in situ CO at Maïdo, using (a) all paired data points, (b) only daytime points and (c) only nighttime points. The red line is the regression line while the black dashed line is the identity line.